# Conformal Tail Risk Control for Large Language Model Alignment

**Catherine Chen** [1]  **Jingyan Shen** [2]  **Zhun Deng** [3]  **Lihua Lei** [4][5]

## Abstract

Recent developments in large language models (LLMs) have led to their widespread usage for various tasks. The prevalence of LLMs in society implores the assurance on the reliability of their performance. In particular, risk-sensitive applications demand meticulous attention to unexpectedly poor outcomes, i.e., tail events, for instance, toxic answers, humiliating language, and offensive outputs. Due to the costly nature of acquiring human annotations, general-purpose scoring models have been created to automate the process of quantifying these tail events. This phenomenon introduces potential human-machine misalignment between the respective scoring mechanisms. In this work, we present a lightweight calibration framework for blackbox models that ensures the alignment of humans and machines with provable guarantees. Our framework provides a rigorous approach to controlling any distortion risk measure that is characterized by a weighted average of quantiles of the loss incurred by the LLM with high confidence. The theoretical foundation of our method relies on the connection between conformal risk control and a traditional family of statistics, i.e., L-statistics. To demonstrate the utility of our framework, we conduct comprehensive experiments that address the issue of human-machine misalignment.

## 1. Introduction

Large Language Models (LLMs) have proven to be pervasive in society with applications across various settings, including those of high sensitivity. While LLMs generally perform quite well, there remains a low probability associated with the event that the models generate undesirable and even catastrophic outputs, including misinformation, malicious use cases, and harmful/toxic comments. Although these responses are relatively rare, each occurrence has the potential to cause significant harm to individuals or even society as a whole.

Quantitative notions of disutility, such as toxicity, are typically based on human annotations that are costly to acquire. To reduce labor cost, general purpose models, for example, Detoxify (Hanu & Unitary team, 2020), have been created to automatically generate disutility measures for LLM outputs. Given the existence of both human and machine assessments, a common issue is the misalignment of the two metrics, which might be caused by distribution shift of human opinions when deploying the model. This is further complicated by the lack of an inherent scale that governs machine scores. While many techniques, such as, Reinforcement Learning from Human Feedback (RLHF), have been proposed to improve alignment, these approaches typically do not provide guarantees regarding the alignment of machine-generated scores. Even when such guarantees exist, they often rely on assumptions that are hard to defend. Moreover, many RLHF approaches require computationally intensive model refitting (Christiano et al., 2017; Ziegler et al., 2019; Casper et al., 2023).

In this work, we address the issue of misalignment through the lens of risk control. In particular, we treat the human-annotated disutility score as the ground-truth risk measure and calibrate the raw outputs generated by the LLM to control certain functionals of the risk distribution at a pre-specified level. Unlike existing methods, our method does not involve model refitting and provides finite-sample guarantees of risk control under no assumptions about the LLM or the underlying data generation process.

The proposed framework substantially generalizes the existing literature in conformal risk control (Bates et al., 2021; Angelopoulos et al., 2021; 2023), which only holds for traditional risk measures characterized by the expectation of a loss function common in supervised learning problems. These risk measures are not suitable for tail risks associated with low probability events. By contrast, other works only

*Equal contribution [1]Institute for Computational and Mathematical Engineering, Stanford University, CA [2]Department of Industrial Engineering and Operations Research, Columbia University, NY [3]Department of Computer Science, UNC Chapel Hill, NC [4]Graduate School of Business, Stanford University, CA [5]Department of Statistics (by courtesy), Stanford University, CA. Correspondence to: Catherine Chen <cyc2152@stanford.edu>.

*Proceedings of the 42$^{nd}$ International Conference on Machine Learning*, Vancouver, Canada. PMLR 267, 2025. Copyright 2025 by the author(s).

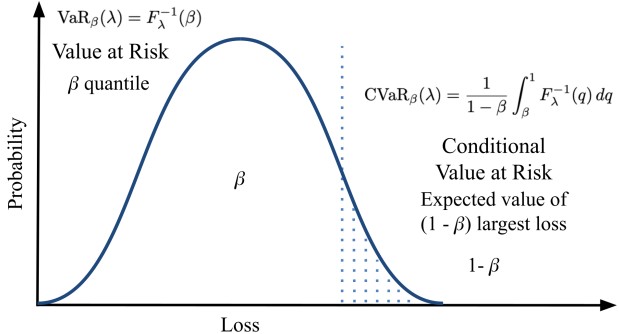

*Figure 1.* **Examples of distortion risk measures: Value-at-Risk (VaR$_\beta$) and Conditional Value-at-Risk (CVaR$_\beta$).**

consider risk measures that are quantiles of a loss function on the human scores of the outputs (Mohri & Hashimoto, 2024; Cherian et al., 2024; Quach et al., 2024). While taking into account the rare events, quantile risk measures often underestimate the tail risk because they do not account for information from the more extreme quantiles. A better suited metric is the Conditional Value-at-Risk (CVaR) (Rockafellar & Uryasev, 2000), which measures the average of a range of upper quantiles. The CVaR is a special case of distortion risk measures, defined as weighted averages of loss quantiles, which is considered by Snell et al. (2022) and Zollo et al. (2023), and shown in (Fig. 1). In this work, we explore how distortion risk control can be applied to align LLMs with respect to any disutility metric and leverage techniques from the theory of L-estimators in statistics (van der Vaart, 1998) to achieve finite-sample control of any distortion risk measure.

## 1.1. Related works and contribution

**Quantile Risk Control.** Traditionally, conformal risk control is characterized by bounding the expected loss of a given predictor to obtain provable guarantees (Bates et al., 2021; Angelopoulos et al., 2021). Recent work has extended the class of loss functions to a wider class of functions defined by the loss for the quantiles of the data distribution, i.e., distortion risks (Snell et al., 2022). Existing methods rely on replacing all quantiles by their confidence envelope to form an *upper confidence bound* (UCB) on the true risk, as described in (Bates et al., 2021). These procedures are presented in more detail in 3.4. In our work, we directly estimate any given distortion risk measure via *L-statistics* to derive tight risk control bounds.

**Connection to Best-of-$N$.** Our work differs from inference-time alignment strategies, such as, best-of-$N$, a fixed-sample inference-time heuristic. In contrast, our method is an adaptive, risk-controlling strategy. In partic-

ular, best-of-$N$ uses a fixed generation count $N$, meaning that it always samples $N$ responses for each prompt. Conversely, our framework uses an adaptive number of samples, $N_x$, depending on the prompt and desired risk level. For example, when a prompt is non-toxic, best-of-$N$ generates $N$ responses, while our method may only require one, yielding a lower average inference cost. We illustrate this difference between the two methods in 4.2. Furthermore, best-of-$N$ implicitly assumes that the disutility function in the LLM framework is well-aligned with the human user. Not only does our method not make such assumptions, but we also extend the problem of alignment by introducing and tuning a parameter that effectively controls the human distuility of an LLM generated response.

## 2. A risk-controlling approach for LLM alignments

### 2.1. Problem setup

An LLM produces a response $y(x) \in \mathcal{Y}$ for any given user prompt $x \in \mathcal{X}$ from a distribution $p(y \mid x)$. The response $y(x)$ could be an answer to a question or a response to a comment made by the user. To evaluate the disutility of $y(x)$, human-annotators are enlisted to rate different aspects of $y(x)$, for instance, misinformation and toxicity. Let $r(y(x))$ denote the human rating of $y(x)$, which is generally random due to cognitive uncertainty. Throughout the paper we assume that $r(y(x)) = 0$ when the LLM declines to respond. An example of a disutility measure is illustrated in the Jigsaw Unintended Bias in Toxicity Classification (cjadams et al., 2019) dataset, which provides toxicity labels from up to 10 human-annotators for each of the 2 million comments on a Civil Comments platform. The total score $r(y(x))$ can be obtained by averaging over the ratings of the annotators. For general disutility metrics, $r(y(x))$ can also be defined as the negative reward estimated using the Bradley-Terry model, as done in RLHF.

In most applications, one can leverage historical data to train a model that estimates the human-rated disutility. For example, Detoxify (Hanu & Unitary team, 2020) is a machine learning model that assesses the toxicity of responses. Denote $r_m(y(x))$ as the machine-generated disutility score for a response $y$. Note that $r_m(y(x))$ typically differs from $r(y(x))$ and may even lack monotonicity with respect to $r(y(x))$. Although machine ratings are inexpensive and scalable, the misalignment, or lack of rank preservation between the machine and human ratings diminishes its reliability. Moreover, it is often hard to interpret the scale of the score and choose the right cut-off to decide whether the generated data should be accepted, especially when applied to different contexts. For example, a sentence with a toxicity score of 0.85 could be very toxic for workplace conversations but acceptable in historian studies. Similarly,

the RLHF reward only captures ordinal preferences rather than cardinal values.

Given any generative model or sampler $p(y \mid x)$, an aligned model seeks to produce an output $\tilde{y}(x)$ in a way such that the overall human-rated disutility, $r(\tilde{y}(x))$, is minimized. We call $\tilde{y}(x)$ a calibrated model.

Let $x$ be a random draw from the population of prompts of interest, and $F_{r(\tilde{y}(x))}$ denote the distribution of $r(\tilde{y}(x))$ over $x$, which depends on the randomness of $\tilde{y}(\cdot)$, and cognitive uncertainty of $r(\cdot)$. To aggregate over different prompts and integrate out the randomness of responses and cognitive uncertainty, we can define a summary measure $R\left(F_{r(\tilde{y}(x))}\right)$ where $F_{r(\tilde{y}(x))}$ denotes the cumulative distribution function (CDF) of $r(\tilde{y}(x))$ with $x$ being a draw from a population of prompts, and $R(\cdot)$ being a functional that maps any distribution to a non-negative number. We denote $R\left(F_{r(\tilde{y}(x))}\right)$ using $R(F)$ as a notational shorthand. As an example, $R(F)$ can be chosen as the mean disutility.

To achieve alignment, we aim to control the risk $R\left(F_{r(\tilde{y}(x))}\right)$ at a pre-specified level $\alpha$. This objective is very different from existing practices (e.g., RLHF) that target the risk associated with the machine disutility rather than the human disutility scores. It is a challenging task because the human rating function $r(\cdot)$ operates as a black box, and researchers can only observe its output for a given set of prompt-response pairs.

Note that any risk level $\alpha$ can be achieved by abstaining from responding to any prompts. Clearly, this should be avoided. As will be seen later, the deployment cost of our calibrated LLM increases as $\alpha$ decreases and could grow to infinity as $\alpha \to 0$, if no abstention is allowed. As a result, though being conservative may appear innocuous, it is unnecessarily costly. To minimize deployment costs, we would want $R\left(F_{r(\tilde{y}(x))}\right)$ to be as close to $\alpha$ as possible.

### 2.2. Choice of risk functions

In classical statistical decision theory, risk is often defined as the expectation of a loss function, i.e. $R(F) = \mathbb{E}_{r \sim F}[L(r)]$, where $L(\cdot)$ is a loss function. However, considering that the majority of generated data is normal, traditional risk measures may fail to capture the disutility as these events are only manifested in the tail.

Instead, we choose $R(F)$ to be a *distortion risk measure* (e.g. Balbás et al., 2009; Snell et al., 2022), defined as a weighted average of quantiles,

$$R_\psi(F) := \int_0^1 F^{-1}(p)\, d\psi \qquad (1)$$

where $F^{-1}(p) \triangleq \inf\{x : F(x) \geq p\}$ denotes the $p$-th quantile of $F$, and $\psi(\cdot)$ is a weighting measure such that

$\psi(p) \geq 0$ and $\int_0^1 d\psi(p) = 1$. An example of a distortion risk measure is the widely-used $\mathrm{CVaR}_\beta$ (Rockafellar & Uryasev, 2000) where $\psi$ is the uniform measure on $[\beta, 1]$. Other examples include mean (with $\psi$ the uniform measure on $[0, 1]$) and $\beta$-th quantile, also known as Value-at-Risk, for any $\beta$ (with $\psi$ the point mass at $p$). Examples of distortion risk measures are shown in (Fig. 1). We also want to remark that expected mean and quantiles are also special cases of distortion risk measures.

### 2.3. Augmentation of LLM outputs

If the original output $y(x)$ does not control the risk at $\alpha$, we need to refine it. Instead of modifying the underlying data generating process $p(y \mid x)$ based on the score by retraining the model, we adopt a light-weight calibration approach that amounts to tuning a one-dimensional parameter instead of the high-dimensional model parameters. Specifically, for every prompt $x$, we request a response from the LLM multiple times to generate a candidate set $\mathcal{C}(x) = \{y_1(x), \ldots, y_N(x)\}$, where $N$ denotes the set size. To maximize the information content, we follow Quach et al. (2024) in eliminating responses to ensure diversity, quality, and set-confidence; see Appendix A.1 for details.

Our idea is to generate a nested sequence of subsets of $\mathcal{C}(x)$ that have increasing disutility. Since we are not allowed to collect human ratings on the fly, we use the machine disutility score as a proxy. In particular, for each $\lambda$ in the range of machine scores $\Lambda$, we generate a set

$$\mathcal{C}_\lambda(x) = \{y(x) \in \mathcal{C}(x) : r_m(y) < \lambda\}$$

by using the machine disutility score $r_m(\cdot)$. Without loss of generality, we assume $\Lambda = [\lambda_{\min}, \lambda_{\max}]$. We assign a disutility score $r_\lambda(x)$ for $C_\lambda(x)$ as the worst human rating, i.e.,

$$r_\lambda(x) = \max_{y \in \mathcal{C}_\lambda(x)} r(y).$$

By design, $r_\lambda(x)$ is non-decreasing in $\lambda$, a key property that our method leverages. Moreover, $\mathcal{C}_{\lambda_{\min}}(x) = \emptyset$ for any $x$ and hence $r_{\lambda_{\min}}(x) = 0$. The process of generating $\mathcal{C}_\lambda(x)$ and $r_\lambda(x)$ is illustrated in Figure 2.

Notably, $\mathcal{C}_\lambda(x)$ can be computed for any $x$ and $\lambda$, because it relies solely on machine disutility scores, whereas $r_\lambda(x)$ is a black-box function that is only available for prompts with collected human ratings. Our goal is to choose $\hat{\lambda}$ based on human-annotated data such that $R(F_{r_{\hat{\lambda}}(x)}) \leq \alpha$. Since $r_{\hat{\lambda}}(x)$ is defined as the worst human score in $\mathcal{C}_{\hat{\lambda}}(x)$, we can pick any candidate $\tilde{y}(x) \in \mathcal{C}_{\hat{\lambda}}(x)$ and the resulting risk $R(F_{r(\tilde{y}(x))})$ will be controlled at level $\alpha$. When $\mathcal{C}_{\hat{\lambda}}(x)$ is empty, the calibrated LLM simply declines to respond. Importantly, the selection can be arbitrary – for example, we could choose the response from $\mathcal{C}_{\hat{\lambda}}(x)$ that has the minimal machine disutility score or maximal information content

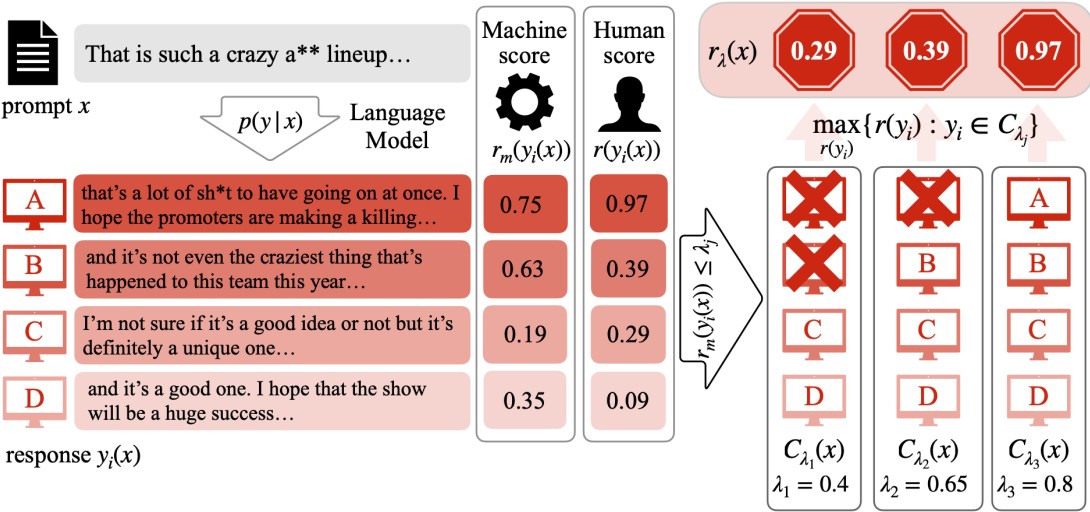

*Figure 2.* **Illustration of the process to generate** $\mathcal{C}(x), \mathcal{C}_\lambda(x),$ **and** $r_\lambda(x)$. We sample $N$ responses $y_i$, $i = 1, \ldots, N$ from the LLM $p(y \mid x)$. Each response is associated with a machine disutility score $r_m(y_i(x))$, and a human-rated disutility score $r(y_i(x))$. To construct $\mathcal{C}_{\lambda_j}(x)$, we keep the responses such that the machine toxicity score satisfies $r_m(y_i(x)) \leq \lambda_j$ for each $\lambda_j \in \Lambda$. Finally, we compute the induced score $r_\lambda(x)$ by taking the maximum human disutility score of each $\mathcal{C}_{\lambda_j}(x)$.

measured by another metric. To summarize, we reduce the task from retraining a calibrated LLM $\tilde{y}(x)$ with high-dimensional model parameters to searching for a univariate parameter $\hat{\lambda}$.

# 3. Distortion risk control via L-statistics

## 3.1. Theoretical setting

Suppose we sample $n$ prompts $x_1, \ldots, x_n$ i.i.d. from a distribution. For each prompt, we generate the candidate set $\mathcal{C}(x_i)$ as described in Section 2.3, then recruit human raters to score all responses in the set. Following the procedure in Section 2.3, we can obtain a dataset $\mathcal{D}$ that includes $(x_i, \{r_\lambda(x_i) : \lambda \in \Lambda\})$ for $i = 1, \ldots, n$. We assume that the machine disutility score model is pretrained and independent of our dataset $\mathcal{D}$. Then the data points $(x_i, \{r_\lambda(x_i)) : \lambda \in \Lambda\})$ remain i.i.d. Throughout the rest of the section we denote a generic draw from the prompt distribution by $x$.

For each $\lambda \in \Lambda$, we use the shorthand notation $R_\psi(\lambda)$ for $R_\psi(F_{r_\lambda(x)})$, where $R_\psi$ is the distortion risk measure defined in (1) with a user-chosen weight measure $\psi$. Our goal is to learn $\hat{\lambda}$ from $\mathcal{D}$ such that

$$\mathbb{P}_\mathcal{D}(R_\psi(\hat{\lambda}) \leq \alpha) \geq 1 - \delta,$$

for some pre-specified $(\alpha, \delta)$. Above, $\mathbb{P}_\mathcal{D}$ accounts for the randomness in $\mathcal{D}$. The parameter $\alpha$ represents the target risk level and $1 - \delta$ corresponds to the confidence level. This formulation aligns with the standard objective of probably approximately correct (PAC) learning and has also been

studied in the context of conformal risk control (Bates et al., 2021; Angelopoulos et al., 2021).

## 3.2. L-statistics

L-statistics refers to the class of estimators expressed as linear combinations of order statistics originating from (Mosteller, 1946). Notable examples include the sample quantile, the trimmed mean, and the winsorized mean (Tukey, 1962). Fixing $\lambda \in \Lambda$, let $r_{\lambda,(1)} \leq r_{\lambda,(2)} \leq \ldots \leq r_{\lambda,(n)}$ denote the ordered statistics of $(r_\lambda(x_1), \ldots, r_\lambda(x_n))$. Furthermore, let $F_\lambda$ and $\hat{F}_{n,\lambda}$ be the true and empirical distributions of $(r_\lambda(x_1), \ldots, r_\lambda(x_n))$, respectively. For a given distortion risk measure, the plug-in estimator that replaces the true distribution $F$ by the empirical distribution $\hat{F}_{n,\lambda}$ is

$$\hat{R}_\psi(\lambda) = R_\psi(\hat{F}_{n,\lambda}) = \int \hat{F}_{n,\lambda}^{-1}(p)d\psi(p). \qquad (2)$$

By definition, $\hat{F}_{n,\lambda}^{-1}(p) = r_{\lambda,(i)}$ for any $p \in \left(\frac{i-1}{n}, \frac{i}{n}\right]$, thus it can be written as an L-statistic

$$\hat{R}_\psi(\lambda) = \sum_{i=1}^{n} \left\{ \psi\left(\frac{i}{n}\right) - \psi\left(\frac{i-1}{n}\right) \right\} r_{\lambda,(i)}.$$

Theorem 22.3 of van der Vaart (1998) describes the asymptotic normality of L-statistics. Given this result, we show that $\hat{R}_\psi(\lambda)$ is an asymptotically normal estimator of $R_\psi(\lambda)$ for any fixed $\lambda \in \Lambda$ and we can find a consistent variance estimator. The proof of Theorem 3.1 is presented in Appendix C.

**Theorem 3.1.** *Assume $r_\lambda(x) \in [a, b]$ almost surely for some $-\infty < a < b < \infty$, $F_\lambda$ is continuous and strictly increasing. Further, assume that $\psi(y) = \int_0^y \psi'(z)dz$ for some $\psi'$ that is bounded and continuous at $F_\lambda(r)$ for Lebesgue almost-every $r$. Then,*

$$\frac{\sqrt{n}(\hat{R}_\psi(\lambda) - R_\psi(\lambda))}{\hat{\sigma}(\lambda)} \xrightarrow{d} N(0, 1),$$

*where*

$$\hat{\sigma}^2(\lambda) = \int \int \psi'(\hat{F}_{n,\lambda}(r))\psi'(\hat{F}_{n,\lambda}(\tilde{r}))D_\lambda(r, \tilde{r}) \, dr \, d\tilde{r},$$

*with $\hat{F}_{n,\lambda}$ being the empirical distribution of $r_\lambda(x_1), \ldots, r_\lambda(x_n)$ and*

$$D_\lambda(r, \tilde{r}) = \hat{F}_{n,\lambda}(r \wedge \tilde{r}) - \hat{F}_{n,\lambda}(r)\hat{F}_{n,\lambda}(\tilde{r}).$$

*Equivalently,*

$$\hat{\sigma}^2(\lambda) = \frac{1}{n^2} \sum_{i,j=1}^{n} \psi'\left(\frac{i}{n}\right) \psi'\left(\frac{j}{n}\right) \left(\frac{i \wedge j}{n} - \frac{ij}{n^2}\right). \quad (3)$$

We assume the boundedness of $r_\lambda(x_i)$ for simplicity. It is a reasonable assumption for ratings which are often designed to be bounded. It can be relaxed with more involved proof techniques (Gardiner & Kumar Sen, 1979; Stigler, 1974).

Among all distortion risk measures, $\mathrm{CVaR}_\beta$ is the most interpretable and widely-used metric. For $\mathrm{CVaR}_\beta$, we can find a much simpler variance estimator.

**Corollary 3.2.** *For $\mathrm{CVaR}_\beta$ with $\psi(p) = \max\{p - \beta, 0\}/(1 - \beta)$, Theorem 3.1 holds with*

$$\hat{\sigma}^2(\lambda) = \frac{1}{(1-\beta)^2}\widehat{\mathrm{Var}}\left(\{\max\{r_\lambda(x_i), r_{\lambda,(\lceil n\beta \rceil)}\}\}_{i=1}^{n}\right),$$

*where $\widehat{\mathrm{Var}}$ denotes the sample variance.*

Another important example is $\mathrm{VaR}_\beta$. While the VaR is not a distortion risk measure with a differentiable $\psi$, we develop parallel theory in Appendix D based on the asymptotic theory of empirical quantiles. Unlike $\mathrm{CVaR}_\beta$, the asymptotic variance depends on the density $F'_\lambda$ of $r_i$ and hence harder to estimate. We apply the bootstrap technique (Efron & Tibshirani, 1994) instead to estimate $\hat{\sigma}^2(\lambda)$.

### 3.3. Conformal distortion risk control via L-statistics

By design, $R_\psi(\lambda)$ is monotonic in $\lambda$. This allows us to apply the method of Bates et al. (2021) to achieve risk control by inverting a pointwise upper confidence bound (UCB). Specifically, we choose

$$\hat{\lambda} = \max\{\lambda \in \Lambda : \hat{R}_\psi^+(\lambda') \leq \alpha, \forall \lambda' \leq \lambda\}, \quad (4)$$

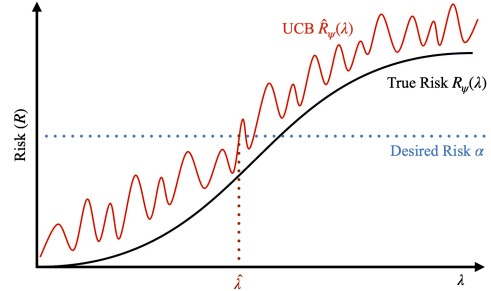

*Figure 3.* **Illustration of** $\hat{\lambda}$. We choose $\hat{\lambda}$ as the last $\lambda$ such that the (asymptotic) upper confidence bound $\hat{R}_\psi^+(\lambda)$ falls below $\alpha$.

where

$$\hat{R}_\psi^+(\lambda) = \hat{R}_\psi(\lambda) + z_{1-\delta} \cdot \hat{\sigma}(\lambda),$$

and $z_{1-\delta}$ is the $(1-\delta)$-th quantile of the standard normal distribution, as illustrated in Fig. 3. In practice, we discretize $\Lambda$ to avoid checking the condition $\hat{R}_\psi^+(\lambda) \leq \alpha$ for infinitely many points. Our procedure is outlined in Algorithm 1 and illustrated in Fig. 2.

By Theorem 3.1, $\hat{R}_\psi^+(\lambda)$ is an asymptotic $(1 - \delta)$ UCB. Although Bates et al. (2021) focus on traditional risk measures that are expressed as expected loss, we can easily extend their result (Theorem 6) to distortion risk measures.

**Theorem 3.3.** *Let $\hat{\lambda}$ be defined in (4). Assume $R_\psi(\lambda)$ is continuous and strictly increasing. In the same setting as Theorem 3.1*

$$\liminf_{n\to\infty} \mathbb{P}_\mathcal{D}\left(R_\psi\left(\hat{\lambda}\right) \leq \alpha\right) \geq 1 - \delta. \quad (5)$$

*As a consequence, for any selection mechanism that picks $\tilde{y}(x)$ from $\mathcal{C}_{\hat{\lambda}}(x)$,*

$$\liminf_{n\to\infty} \mathbb{P}_\mathcal{D}\left(R_\psi\left(F_{r(\tilde{y}(x))}\right) \leq \alpha\right) \geq 1 - \delta.$$

Remarkably, this result does not involve any assumption on the underlying LLM $p(y \mid x)$ or machine disutility score model $r_m(x)$. In particular, it allows the machine ratings to be arbitrarily misaligned with human ratings.

### 3.4. Alternative (conservative) approaches for distortion risk control

Another strategy to construct pointwise UCBs for $R_\psi(\lambda)$ is to replace all quantiles by their confidence envelopes. Specifically, if we have statistics $\{\hat{q}_p^+(\lambda) : p \in [0, 1]\}$ such that

$$\mathbb{P}(q_p(\lambda) \leq \hat{q}_p^+(\lambda), \; \forall p \in [0, 1]) \geq 1 - \delta, \quad (6)$$

then $\int_0^1 \hat{q}_p^+(\lambda)d\psi(p)$ is a valid $(1 - \delta)$ UCB for $R_\psi(\lambda)$. Following Snell et al. (2022), we consider two confidence envelopes based on the Dvoretzky-Kiefer-Wolfowitz (DKW) inequality and Berk-Jones (BJ) statistics, respectively. For simplicity, we assume $F_\lambda$ is continuous.

**Algorithm 1** Conformal distortion risk control

---

**Input:** machine-scoring model $r_m(\cdot)$, discrete subset of its range $\Lambda$, human-annotated scores $r(\cdot)$, set of prompts $\mathcal{X} = (x_i)_{i=1}^n$, target level $\alpha$, tolerance level $\delta$, weighting function $\psi$.

**function** DRC($\mathcal{X}, \Lambda, r_m(\cdot), r(\cdot), \alpha, \delta, \psi$)

**for** $x_i \in \mathcal{X}$ **do**

  $\mathcal{C} \leftarrow$ CANDIDATESET($x_i$) {See Algorithm 2}

  **for** $\lambda \in \Lambda$ **do**

    $\mathcal{C}_\lambda(x_i) = \{\}$

    **for** $y_k \in \mathcal{C}(x_i)$ **do**

      **if** $r_m(y_k) < \lambda$ **then**

        $\mathcal{C}_\lambda(x_i) \leftarrow \mathcal{C}_\lambda(x_i) \cup \{y_k\}$

      **end if**

      $r_\lambda(x_i) \leftarrow \max\{r(y_j) : y_j \in \mathcal{C}_\lambda(x_i)\}$

    **end for**

  **end for**

**end for**

**for** $\lambda \in \Lambda$ **do**

  $(r_{\lambda,(1)}, \ldots, r_{\lambda,(n)}) \leftarrow$ SORT($r_\lambda(x_1), \ldots, r_\lambda(x_n)$)

  $\hat{R}_\psi(\lambda) \leftarrow \sum_{i=1}^n \left\{ \psi\left(\frac{i}{n}\right) - \psi\left(\frac{i-1}{n}\right) \right\} r_{\lambda,(i)}$

  $\hat{\sigma}^2(\lambda) \leftarrow$ Eq. (3)

  $\hat{R}_\psi^+(\lambda) \leftarrow \hat{R}_\psi(\lambda) + z_{1-\delta} \cdot \frac{\hat{\sigma}(\lambda)}{\sqrt{n}}$ {UCB}

**end for**

$\hat{\lambda} \leftarrow \max\left\{ \lambda \in \Lambda : \hat{R}_\psi^+(\lambda') \leq \alpha, \forall \lambda' \leq \lambda \right\}$

**return** $\hat{\lambda}$

---

**DKW inequality.** The DKW inequality implies that, for any $\lambda \in \Lambda$ and $\epsilon > 0$,

$$\mathbb{P}\left( \sup_{r \in \mathbb{R}} | \hat{F}_{n,\lambda}(r) - F_\lambda(r) | \geq \epsilon \right) \leq 2e^{-2n\epsilon^2}.$$

Letting $\epsilon_{n,\delta} = \sqrt{\log(2/\delta)/2n}$ and $r = q_p(\lambda)$ for $p \in [0,1]$, we obtain that, with probability at least $1 - \delta$,

$$|\hat{F}_{n,\lambda}(q_p(\lambda)) - p| \leq \epsilon_{n,\delta}, \quad \forall p \in [0,1].$$

This implies an upper confidence envelope as $\hat{q}_p^+(\lambda) = r_{\lambda,(k_{n,p})}$, where $k_{n,p} = \lceil n(p + \epsilon_{n,\delta}) \rceil$. Here, if $k_{n,p} > n$, we set $r_{\lambda,(k_{n,p})} = \lambda_{\max}$, the upper bound of $\Lambda$.

**Berk-Jones Statistics.** As pointed out by Snell et al. (2022) and Bates et al. (2023), the DKW inequality is overly conservative for $p$ close to 0 and 1. The BJ statistic uses the fact that $F_r(\lambda_{r,(i)}) \sim \text{Beta}(i, n - i + 1)$, where Beta denotes a Beta-distribution. Let $B_{i,n-i+1}$ denote the cumulative distribution function of $\text{Beta}(i, n - i + 1)$. The BJ statistic is then defined as

$$M_n^+ = \max_{1 \leq i \leq n} G_{i,n-i+1}(F_r(r_{\lambda,(i)})).$$

Let $s_\delta$ be the $\delta$-th quantile of $M_n^+$. Clearly, $s_\delta$ does not depend on $F_r$ because $F_r(r_{\lambda,i}) \sim \text{Unif}([0,1])$ when $F_r$ is con-

tinuous. It can be computed in polynomial time (Moscovich & Nadler, 2017). Let $s_i = G_{i,n-i+1}^{-1}(s_\delta)$. Then

$$\mathbb{P}(q_{s_i}(\lambda) \leq r_{\lambda,(i)}, \ \forall i \leq n) \geq 1 - \delta.$$

This yields an upper confidence envelope $\hat{q}_p^+(\lambda) = r_{\lambda,(i)}$ for any $p \in (s_{i-1}, s_i]$. For $p > s_n$, we set $\hat{q}_p^+(\lambda) = \lambda_{\max}$.

While both DKW and BJ approaches yield finite-sample valid UCBs for $R_\psi(\lambda)$, they are conservative because they do not target the specific choice of $\psi$. In fact, (6) implies that $\int_0^1 \hat{q}_p^+(\lambda) d\psi(p)$ is a uniform UCB across all weight functions, i.e.,

$$\mathbb{P}\left( R_\psi(\lambda) \leq \int_0^1 \hat{q}_p^+(\lambda) d\psi(p), \ \forall \psi \right) \geq 1 - \delta.$$

Therefore, the actual coverage (i.e., probability that $R_\psi(\lambda)$ is less than or equal to the above UCB) is typically much higher than $1 - \delta$. By contrast, the UCB given by the L-statistic is tailored to $R_\psi(\lambda)$ and the coverage converges to $1 - \delta$ as $n \to \infty$.

### 3.5. Deployment of the calibrated model

Recall that any choice of $\tilde{y}(x) \in \mathcal{C}_{\hat{\lambda}}(x)$ controls the risk $R_\psi(F_{\tilde{y}(x)})$. For a new prompt $x$, the most cost-effective approach is to sample candidate responses $y_1, y_2, \ldots$ from the underlying LLM $p(y \mid x)$ until the first time the machine disutility score is below $\hat{\lambda}$. Suppose each sample incurs a unit of computational cost. Given a prompt $x$, the sampling cost follows a geometric distribution with rate $\mathbb{P}(r_m(y) < \hat{\lambda} \mid x, \hat{\lambda})$. Therefore, on average, for a given prompt $x$, we expect the number of samples needed to generate a response with disutility score less than $\hat{\lambda}$ is

$$N_x = \frac{1}{\mathbb{P}(r_m(y) < \hat{\lambda} \mid x, \hat{\lambda})}.$$

This demonstrates that the number of responses generated by our method, CDRC-L, is adaptive to the given prompt $x$, unlike best-of-$N$. In other words, the (unconditional) expected cost is

$$\text{Cost}(\hat{\lambda}) = \mathbb{E}_x \left[ \frac{1}{\mathbb{P}(r_m(y) < \hat{\lambda} \mid x, \hat{\lambda})} \right].$$

Suppose we have a hold-out set of prompts $\mathcal{D}'$ that is independent of the dataset $\mathcal{D}$ defined in Section 3.1, $\text{Cost}(\hat{\lambda})$ can be estimated by

$$\frac{1}{|\mathcal{D}'|} \sum_{x \in \mathcal{D}'} \frac{1}{\widehat{\mathbb{P}}(r_m(y) < \hat{\lambda} \mid x, \hat{\lambda})}, \tag{7}$$

where the probability is estimated by the Monte-Carlo method.

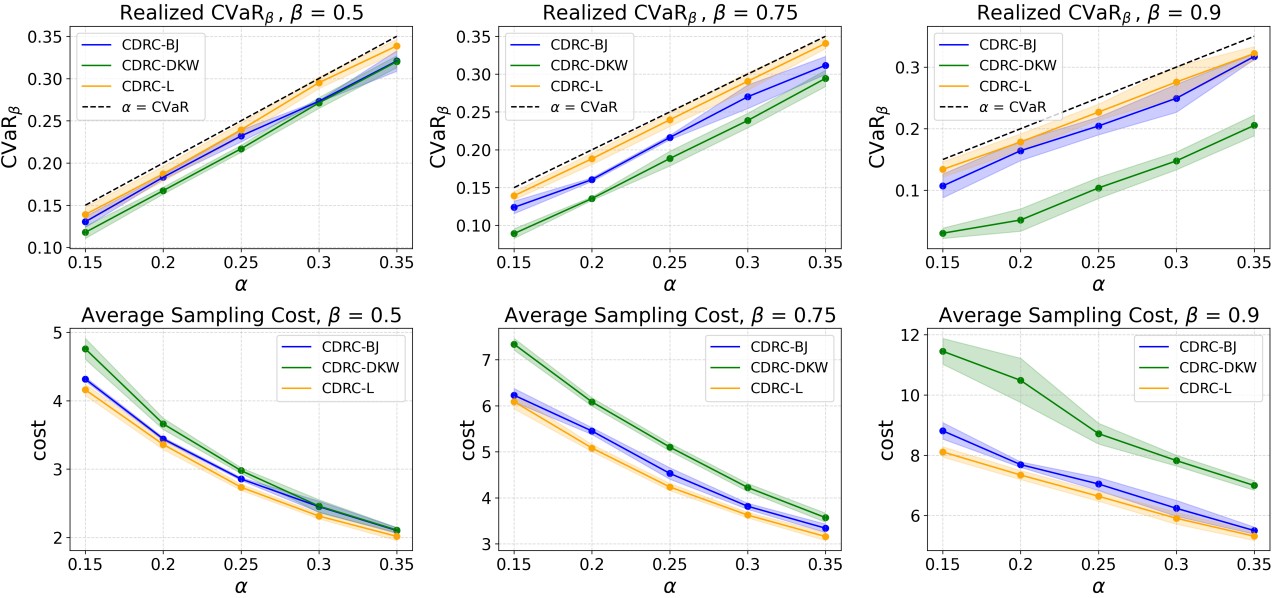

*Figure 4.* **LLaMA2-7B: Realized** $\mathrm{CVaR}_\beta$ **vs.** $\alpha$ **(row 1) and average sampling cost vs.** $\alpha$ **(row 2) for Spearman correlation between human and machine toxicity scores at** $\rho = 0.57$ **evaluated on held-out dataset with** $|\mathcal{D}| = 6000$. The confidence band is computed by taking the mean estimate plus/minus one standard error estimated from the results across independent experiments. Each subplot in the respective rows illustrates a different setting of $\beta \in \{0.5, 0.75, 0.9\}$. From the panels in the first row, we observe that our method, CDRC-L (orange), is an improvement to CDRC-DKW (green) and CDRC-BJ (blue), as it is able to achieve risk control (shown in the black dotted line) while being less conservative than both baseline methods. Evident from the panels in the second row, our method is more cost efficient and least conservative in generating a risk-controlled LLM response than DKW or BJ.

## 4. Experiments

In this section, we perform experiments to investigate the issue of human-machine misalignment by implementing our conformal distortion risk control method to mitigate toxicity of LLM-generated reseponses. This is a critical application, as toxic outputs may cause severely negative impacts on impressionable populations, moreover, propagate across wide audiences, leading to misinformation and harm.

### 4.1. Experimental setup

**Datasets and models.** We randomly draw 10K prompts from the REALTOXICITYPROMPTS dataset (Gehman et al., 2020) . For each selected prompt $x_i$, we generate 40 responses $y_j(x_i)$ using the LLaMA2-7B model (Touvron et al., 2023). Given the initial responses, we apply the sequential algorithm described in Algorithm 2 to construct the candidate response sets $\mathcal{C}(x_i)$, ensuring the quality of the selected responses. Specifically, we use perplexity (PPL) to evaluate response quality, ROUGE-L to assess similarity between responses, and restrict the maximum set size to 32 as a stopping criterion. More details can be found in Appendix A.1.

**Toxicity scores.** To apply our method, we need a human toxicity score function $r(\cdot)$ and a machine toxicity score function $r_m(\cdot)$. Human-annotated data can be costly and time-consuming to acquire. To evaluate our method, we create a cheap semi-synthetic benchmark using an existing machine scoring model as the "human annotator," and a biased model as the "machine assessor." Specifically, we use the Detoxify model (Hanu & Unitary team, 2020) for $r(\cdot)$ and retrain the Detoxify model for $r_m(\cdot)$ on a biased subset of the Jigsaw Unintended Bias in Toxicity Classification dataset (cjadams et al., 2019) that consists of the $c\%$ most and least toxic instances. The goal is to design $r_m(\cdot)$ with varying degrees of misalignment from $r(\cdot)$. This allows us to study the effect of misalignment on $\hat{\lambda}$ and hence the cost of tail risk control. In particular, we quantify the misalignment between human and machine scoring models by the Spearman correlation coefficient $\rho$ between the scores across all candidate responses. In our experiment, we train three models for $r_m(\cdot)$ with $c\% \in \{15\%, 30\%, 70\%\}$. The Spearman correlation coefficients are $0.57, 0.68, 0.78$, respectively. More details about these models can be found in Appendix A.2.1.

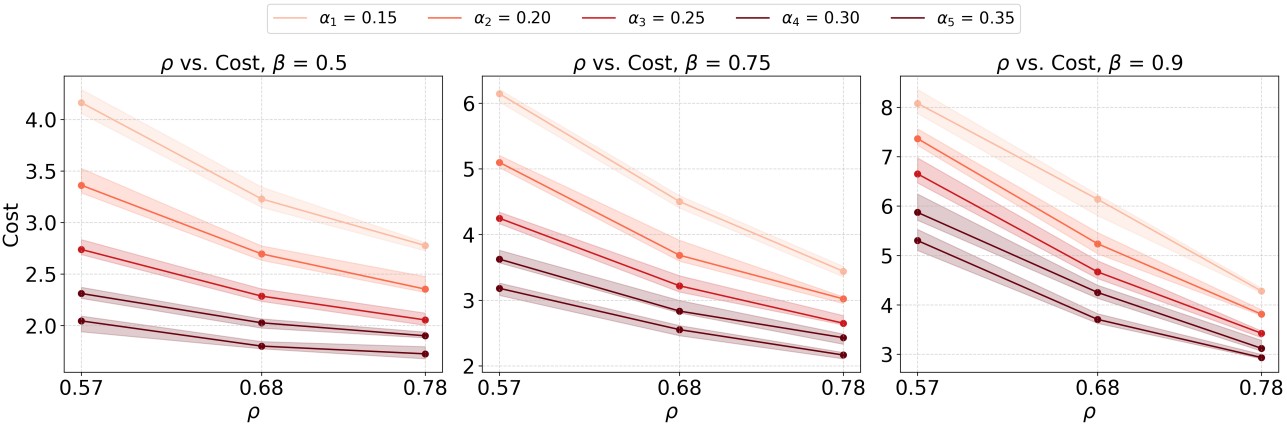

*Figure 5.* **LLaMA2-7B: Spearman correlation vs. average sampling cost for** $\mathrm{CVaR}_\beta$ with $|\mathcal{D}| = 6000$.

**Choices of parameters.** We consider both $\mathrm{CVaR}_\beta$ and $\mathrm{VaR}_\beta$ control with $\beta \in \{0.5, 0.75, 0.9\}$. We fix the confidence parameter $1 - \delta = 0.95$. To determine a reasonable target level $\alpha$, we compute the empirical $\mathrm{CVaR}_q$ on human scores of all candidate responses with $q \in \{1\%, 5\%, 10\%, 15\%, 20\%\}$; see Appendix A.3. This suggests a range of reasonable target levels. In particular, we consider $\alpha \in \{0.15, 0.2, 0.25, 0.3, 0.35\}$.

**Evaluation.** We randomly split the prompts, using $|\mathcal{D}| \in \{50, 100, 200, 1000, 6000\}$ to determine the optimal threshold $\hat{\lambda}$ and the remaining as a held-out test dataset. For each method, after selecting $\hat{\lambda}$, we deploy the calibrated model on the held-out dataset. We then evaluate the realized $\mathrm{CVaR}_\beta$ and $\mathrm{VaR}_\beta$ of human scores and estimate the sampling cost following (7). We apply all three versions of conformal distortion risk control based on L-statistics, DKW inequality, and BJ statistics. We refer to them as CDRC-L, CDRC-DKW, and CDRC-BJ, respectively, where CDRC stands for conformal distortion risk control. For CDRC-L and CDRC-DKW, we repeat for 15 times, and for CDRC-BJ, we repeat for 3 times due to computational complexity to compute $s_\delta$.

**Additional models.** We conduct additional experiments using the LLaMA3.2-3B, and LLaMA3.1-8B models. Following the same implementation details as described, we obtain results that are highlighted in Appendix E.

### 4.2. Results

**Realized risk and average cost analysis.** Fig. 4 shows the realized $\mathrm{CVaR}_\beta$ of human scores and the average sampling cost with $\rho = 0.57$ on the held-out dataset as functions of $\alpha$ for $\beta \in \{0.5, 0.75, 0.9\}$ with $|\mathcal{D}| = 6000$. The panels in the first row shows that all methods control the risk at the target level and CDRC-L is least conservative, as dis-

cussed in Section 3.4. As a result, it incurs the smallest deployment cost among all three methods. Moreover, as $\beta$ increases, the advantage of CDRC-L is more prominent. Although CDRC-BJ improves upon CDRC-DKW due to the tighter bounds for extreme quantiles, it still underperforms CRDC-L. On the other hand, the gap between the target and realized $\mathrm{CVaR}_\beta$ stays nearly constant for CDRC-L across different values of $\beta$, suggesting that L-statistics are adaptive to different choices of $\psi$. Results for $\rho = 0.68$, and $\rho = 0.78$ are presented in Appendix B.1, showing similar patterns.

We present the results for $\mathrm{VaR}_\beta$ control with $\rho \in \{0.57, 0.68, 0.78\}$ in Appendix D. The comparison between CDRC-L, CDRC-DKW, and CDRC-BJ are qualitatively similar to $\mathrm{CVaR}_\beta$ control. Notably, the realized $\mathrm{VaR}_\beta$ of CDRC-L is extremely close to the target level with low sampling cost, demonstrating that it is not conservative while remaining the most cost effect among all methods.

**Cost and misalignment.** Next, we examine how the deployment cost varies with the misalignment between human and machine ratings. Fig. 5 demonstrates that, for all settings of $\beta$, as the Spearman correlation coefficient increases, the cost of generating a $\mathrm{CVaR}_\beta$-controlled LLM response drops. This confirms our intuition that better-aligned machine ratings reduces the cost of calibration. Similar trends are observed for $\mathrm{VaR}_\beta$ control, as shown in Appendix D.

**Performance in small samples.** Human annotations are the primary contributors to the calibration cost in our framework. Nevertheless, we demonstrate that our method retains statistical efficiency even with smaller calibration sets, such as, $|\mathcal{D}| \in \{50, 100, 200, 1000\}$

From Fig. 6, it is evident that CDRC-L is consistently the least conservative among all methods, while still maintain-

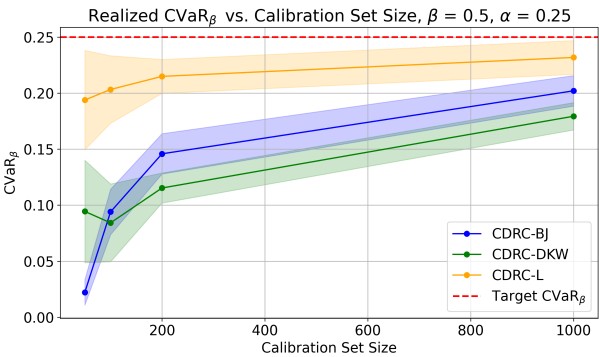

*Figure 6.* **LLaMA2-7B: Realized** $\text{CVaR}_\beta$ **vs. calibration set size** for $\beta = 0.5$, $\alpha = 0.25$, $\rho = 0.57$. The confidence bands are constructed by taking the mean estimate $\pm$ one standard error estimated from the results across 15 independent experiments.

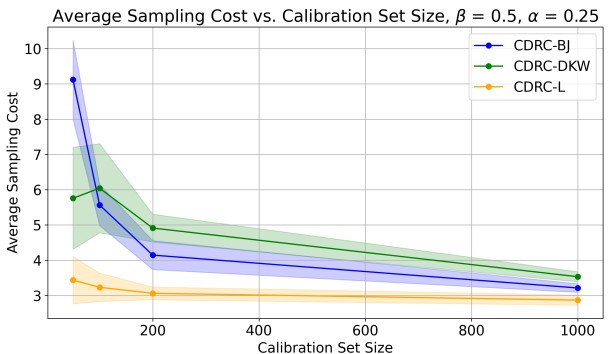

*Figure 7.* **LLaMA2-7B: Average sampling cost for** $\text{CVaR}_\beta$ **control vs. calibration set size** for $\beta = 0.5$, $\alpha = 0.25$, $\rho = 0.57$. The confidence bands are constructed by taking the mean estimate $\pm$ one standard error estimated from the results across 15 independent experiments.

ing the guaranteed $\text{CVaR}_\beta$ control. Furthermore, from Fig. 7, we observe that CDRC-L is consistently the most cost effective among all methods. In particular, it is significantly less expensive in comparison to other methods in settings with small calibration set sizes. We show qualitatively equivalent results for $\text{VaR}_\beta$ in Appendix D.

**Comparison to best-of-$N$.** We compared the two methods under a controlled setup: setting $\beta = 0.5$, using $N \in 3, 5$ for best-of-$N$, and calibration set size $n = 1000$. Following the procedure in 4.1, we evaluated the realized $\text{CVaR}_\beta$ of human-annotated toxicity scores using the response with the lowest machine score from each group of $N$ samples:

| Best-of-N | Mean $\text{CVaR}_\beta \pm$ standard deviation |
|---|---|
| Best-of-3 | $0.2427 \pm 0.0068$ |
| Best-of-5 | $0.2015 \pm 0.007$ |

*Table 1.* Beta $\text{CVaR}_\beta$ Human Scores with standard deviation across trials.

The results in Table 1 demonstrate that our method achieves comparable $\text{CVaR}_\beta$ control with lower average inference cost than best-of-$N$ (see Fig. 4). In particular, on average, for $\beta = 0.5$, best-of-3 controls $\text{CVaR}_\beta$ at approximately level $\alpha = 0.25$, and CDRC-L can achieve the same level of control by generating less than 3 LLM responses. Furthermore, on average, for $\beta = 0.5$, best-of-5 controls $\text{CVaR}_\beta$ at approximately level $\alpha = 0.20$, while CDRC-L can achieve the same level of control by generating less than 3.5 responses.

## 5. Discussion

We close the paper by discussing a few potential generalizations and extensions of our method.

Theorem 3.3 only works for the population of prompts that has the same distribution as the data the method operates on. In the presence of distribution shifts, we can potentially adapt the reweighting technique for traditional expected risk measures (Tibshirani et al., 2019; Lei & Candès, 2021; Candès et al., 2023) to correct for bias for L-statistics.

Due to the rise of RLHF, preference data has become increasingly prevalent. While the above method only works with direct human ratings, there is a potential of adaptation to handle preference data, which exhibits a U-statistic structure.

## Impact Statement

This paper presents work whose goal is to advance the field of machine learning by ensuring safe deployment and alignment of large language models (LLMs). There are many potential societal consequences of our work, for instance, addressing the potential misalignment between human and machines, recalibrating machines such that they are better aligned with humans, and controlling the tail risks associated with language models. In this project, we did not use directly use data from humans. There are not negative impacts which we feel must be specifically highlighted here.

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

## Supplementary Materials

In the supplementary materials, we provide implementation details, additional experimental results, and formalized proofs. Code repository can be found at https://github.com/jy-evangeline/DRC.

## Notation

*Table 2.* Summary of notation used in the paper.

| Symbol | Meaning |
|---|---|
| $\mathcal{X}$ | User prompt space. |
| $\mathcal{Y}$ | LLM output space. |
| $x$ | User prompt drawn from $\mathcal{X}$. |
| $y(x)$ | LLM response given a user prompt $x \in \mathcal{X}$. |
| $\tilde{y}(x)$ | Calibrated LLM response given a user prompt $x \in \mathcal{X}$. |
| $p(x \mid y)$ | Data generating process of LLM. |
| $\mathcal{C}(x)$ | Set of uncalibrated responses generated by an LLM, where $|\mathcal{C}(x)| = N$ for a given prompt $x$. |
| $\mathcal{C}_\lambda(x)$ | Set of responses $y$ generated by an LLM satisfying $r_m(y) < \lambda$ for a given prompt $x$. |
| $r(y)$ | Human-generated disutility score of response $y(x)$. |
| $r_m(y)$ | Machine-generated disutility score of response $y(x)$. |
| $r_\lambda(x)$ | Disutility score for $\mathcal{C}_\lambda(x)$ as the worst human rating, i.e. $r_\lambda(x) = \max_{y \in \mathcal{C}_\lambda(x)} r(y)$. |
| $F_{r(\tilde{y}(x))}$ | Cumulative distribution function (CDF) of $r(\tilde{y}(x))$ over $x$. |
| $R(F_{r(\tilde{y}(x))})$ | Summary measure of $F_{r(\tilde{y}(x))}$, where $R(\cdot)$ is a functional that maps any distribution to a non-negative number. Denote as $R(F)$ as a shorthand. |
| $\Lambda$ | Range of machine scores with $\Lambda = [\lambda_{\min}, \lambda_{\max}]$ |
| $\mathcal{D}$ | Dataset of $(x_i, \{r_\lambda(x_i) : \lambda \in \Lambda\})$ for $n$ i.i.d. prompts $x_1, \ldots, x_n$. |
| $R_\psi(F_{r(\tilde{y}(x))})$ | Distortion risk measure, where $\psi(\cdot)$ is a weighting measure such that $\psi(p) \geq 0$, and $\int_0^1 d\psi(p) = 1$. Denote as $R_\psi(\lambda)$ as a shorthand. |
| $F_\lambda$ | True distribution of $(r_\lambda(x_1), \ldots, r_\lambda(x_n))$. |
| $\hat{F}_{n,\lambda}$ | Empirical distribution of $(r_\lambda(x_1), \ldots, r_\lambda(x_n))$. |
| $\hat{R}_\psi(\lambda)$ | Empirical distortion risk measure of $(r_\lambda(x_1), \ldots, r_\lambda(x_n))$ with $\hat{R}_\psi(\lambda) = \hat{R}_\psi(\hat{F}_{n,\lambda}) = \int \hat{F}_{n,\lambda}^{-1}(p) d\psi(p)$. |
| $\alpha$ | Prespecified risk control level. |
| $1 - \delta$ | Prespecified confidence level. |

## A. Implementation Details

### A.1. Algorithm to generate the candidate set $\mathcal{C}$

Following the original implementation of LLAMA-2-7B-HF model, we set the generation temperature at 0.8 and the top-p parameter at 0.95. We generate candidate response sets for our DRC framework using Algorithm 2, following Quach et al. (2024). To ensure quality and diversity of generated candidates, we filter responses by retaining only those with a Perpexity less than 2.61 while ensuring that the ROUGE-L scores between samples in the candidate set is not greater than 0.26. Table 3 presents the percentiles of these metrics across all generated responses. Fig. 8 outlines the cardinality of the sets generated by various combinations of $\gamma$.

*Table 3.* Percentiles of different hyperparameters.

| Percentile | 50 | 75 | 80 | 85 | 90 | 95 |
|---|---|---|---|---|---|---|
| ROUGE-L | 0.26 | 0.39 | 0.43 | 0.47 | 0.52 | 0.59 |
| Perplexity | 2.61 | 3.07 | 3.18 | 3.31 | 3.48 | 3.76 |

**Algorithm 2** Generation of the candidate set $\mathcal{C}(x)$

**Input:** input prompt $x$, set-based confidence function $\mathcal{F}$, text similarity function $\mathcal{S}$, sample quality estimator $\mathcal{Q}$, fixed threshold configuration $\gamma = (\gamma_1, \gamma_2, \gamma_3)$, sampling budget $k_{\max}$, with conditional output $p_\theta(y \mid x)$ from a generative model.

**function** CANDIDATESET$(x, \mathcal{F}, \mathcal{S}, \mathcal{Q}, \gamma, k_{\max})$

$\mathcal{C} = \{\}$

**for** $k = 1$ **to** $k_{max}$ **do**

    $y_k \leftarrow y \sim p_\theta(y \mid x)$ {Sample from generative model}

    **if** $\mathcal{Q}(x, y_k) < \gamma_1$ and $\max\{\mathcal{S}(y_k, y_j) : y_j \in \mathcal{C}\} > \gamma_2$ **then**

        $\mathcal{C} \leftarrow \mathcal{C} \cup \{y_k\}$ {Quality and similarity control}

    **end if**

    **if** $\mathcal{F}(\mathcal{C}) \geq \gamma_3$ **then**

        **break**{Set-based confidence guarantee}

    **end if**

**end for**

**return** $\mathcal{C}$

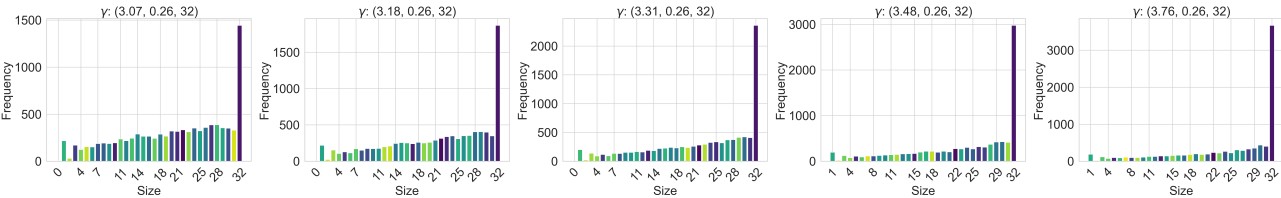

*Figure 8.* **Size of generated sets under different combinations of hyperparameters.** $\gamma = (\gamma_1, \gamma_2, \gamma_3)$ refers to (preplexity, similarity, stopping threshold), respectively.

## A.2. Detoxify model for $r(\cdot)$ and $r_m(\cdot)$

We use the original Detoxify model as a proxy for human annotator scores, then finetune this base model with various sample sizes, a learning rate of $0.0001$, a batch size of $16$, and a weight decay of $3 \times 10^{-6}$ on a single Nvidia A40 GPU. The Adam optimizer was employed with $\alpha = 0.9$, $\beta = 0.999$, and $\epsilon = 10^{-8}$. We exclusively use the Detoxify framework to evaluate the text generated by our model without the prompts.

### A.2.1. A SEMI-SYNTHETIC EXERCISE RESULTS

To study the effect of human-machine misalignment, we create a semi-synthetic data set of human and machine-generated toxicity scores. We use the Detoxify (Hanu & Unitary team, 2020) model as our "human" annotator, and a Detoxify model finetuned on a biased subsample of toxic instances as our "machine" annotator. For details about implementation, see Section 4.1. Table 4 outlines ROC-AUC (Area Under ROC Curve) comparison between the original Detoxify model, and the finetuned model, which illustrates that the original Detoxify model outperforms the finetuned model in predicting toxicity. Fig. 9 illustrates the distribution of human and machine-assigned scores for responses generated by the LLaMA2-7B model across all sampled prompts. Both human and machine toxicity scores exhibit a long-tailed distribution in the real-world dataset. Due to the Detoxify model being fine-tuned with an emphasis on "severe toxicity" samples, its scores are predominantly concentrated in the lower range, highlighting a misalignment between human and machine assessments.

*Table 4.* Performance comparison of detoxify models.

| Model | ROC-AUC |
|---|---|
| Detoxify (original) | 0.97 |
| Finetuned Detoxify ($\rho = 0.57$) | 0.86 |

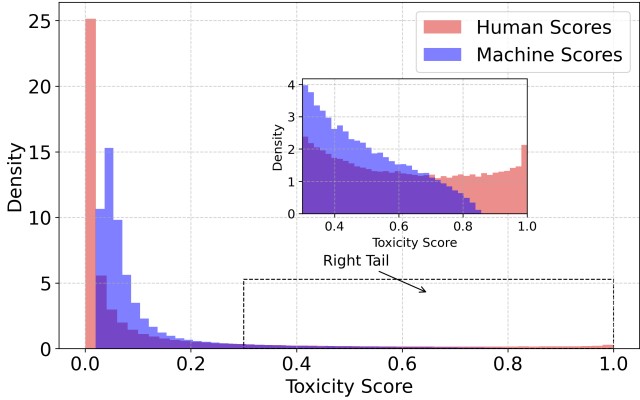

*Figure 9.* Distribution of toxicity scores assigned by humans (i.e., original Detoxify model) and machine (i.e., fine-tuned Detoxify model) for responses generated by the LLaMA2-7B model across all prompts. The main plot shows the density of scores, highlighting a long-tailed distribution in both human and machine evaluations. The zoomed-in plot illustrates that human scores have a heavier tail than machine scores.

### A.3. Choices of $\alpha$

To select an appropriate value of $\alpha$, we arrange $r_\lambda(x)$ in increasing order then rule out the top $q\%$ responses, and compute the desired distortion risk of the lower $(1 - q)\%$. The calculated value is used as the target $\alpha$. For details about implementation, see Section 4.1. Table 5 outlines the rational $\alpha$ values under different settings of $\lambda$ for different $q\%$ given that we are interested in studying CVaR.

*Table 5.* Choice of $\alpha$ with different $q\%$ under various settings of $\gamma$ for CVaR.

| $\boldsymbol{\gamma} = (\gamma_1, \gamma_2, \gamma_3)$ | $q = 1\%$ | $q = 5\%$ | $q = 10\%$ | $q = 15\%$ | $q = 20\%$ |
|---|---|---|---|---|---|
| (3.07, 0.26, 32) | 0.815 | 0.580 | 0.356 | 0.217 | 0.145 |
| (3.18, 0.26, 32) | 0.809 | 0.578 | 0.356 | 0.216 | 0.140 |
| (3.31, 0.26, 32) | 0.815 | 0.574 | 0.354 | 0.219 | 0.142 |
| (3.48, 0.26, 32) | 0.807 | 0.578 | 0.359 | 0.217 | 0.141 |
| (3.76, 0.26, 32) | 0.815 | 0.589 | 0.352 | 0.221 | 0.141 |

## B. Additional experimental results for $\mathrm{CVaR}_\beta$ control

### B.1. Impact of Human-Machine Misalignment

We evaluate the performance, i.e., average sampling cost, and realized $\mathrm{CVaR}_\beta$ on a held-out dataset, of our framework CDRC-L, against baseline methods CDRC-DKW and CDRC-BJ. Fig. 4, Fig. 10, and Fig. 11 illustrate the setting when the Spearman correlation ($\rho$) between the human and machine scores is 0.57, 0.68, and 0.78, respectively. From these figures, it is evident that all methods effectively control the risk at the specified level $\alpha$, given that they all fall below the black dotted line (where $\alpha = \mathrm{CVaR}_\beta$). Our method, CDRC-L, is consistently the least conservative of all methods across different values of $\beta$, and $\rho$. Conversely, baseline methods appear to grow increasingly conservative as $\beta$ increases. As a consequence, we see that our method is consistently the least costly across all settings of $\beta$, and $\rho$.

## C. Proofs

### C.1. Proof of Theorem C.1

We decompose the proof into two parts.

**Theorem C.1** (Asymptotic normality of L-statistics)**.** *Let $f_\lambda, F_\lambda$ denote the PDF and CDF of the score $r_i$, respectively, and*

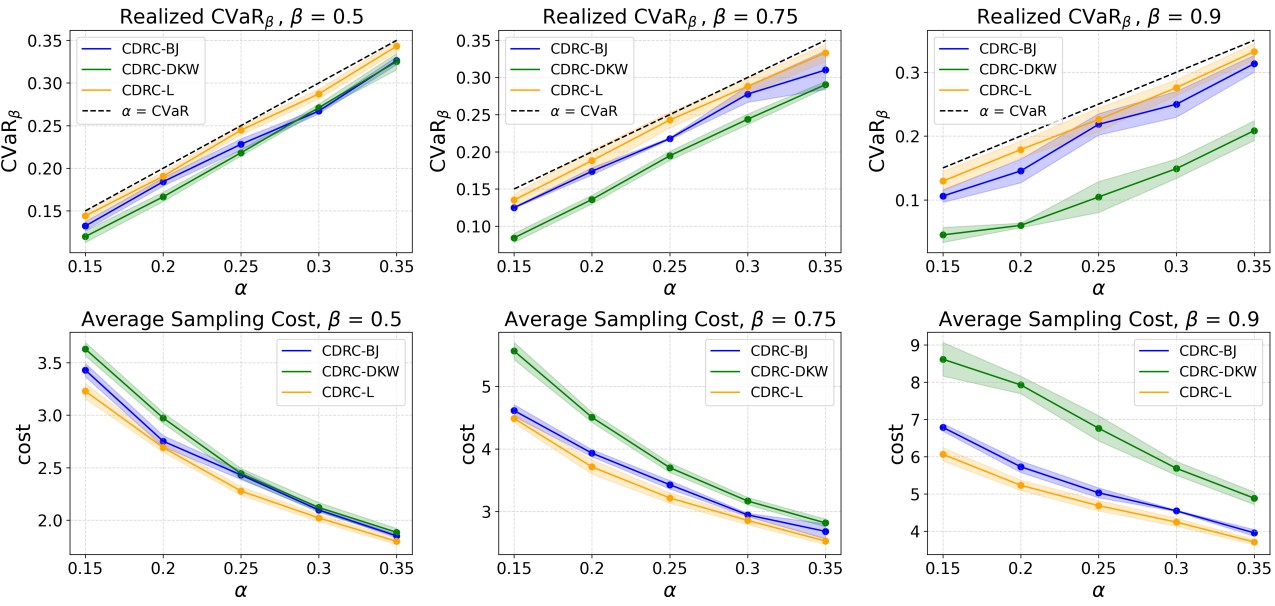

*Figure 10.* **Realized** $\mathrm{CVaR}_\beta$ **and average sampling cost on held-out dataset for** $\rho = 0.68$ with $|\mathcal{D}| = 6000$.

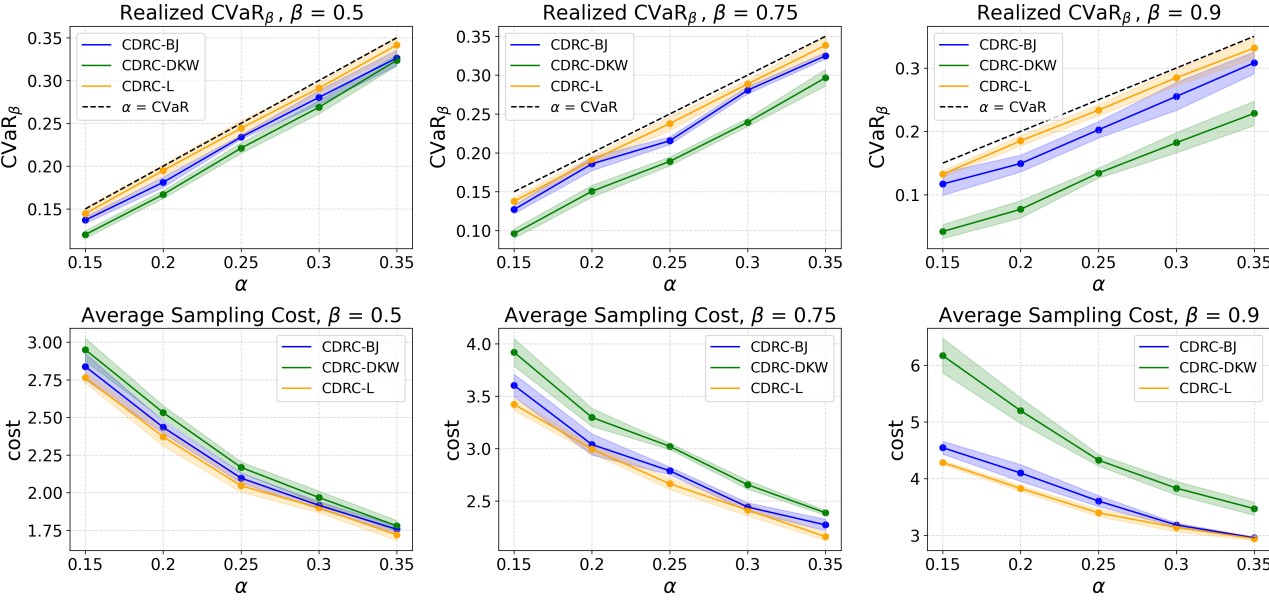

*Figure 11.* **Realized** $\mathrm{CVaR}_\beta$ **and average sampling cost on held-out dataset for** $\rho = 0.78$ with $|\mathcal{D}| = 6000$.

$\hat{F}_{n,\lambda}$ *denote the empirical CDF of the scores* $r_1, \ldots, r_n$. *Further, let*

$$R_\psi(\lambda) = \int_0^1 F_\lambda^{-1}(p)d\psi(p) \quad and \quad \hat{R}_\psi(\lambda) = \int \hat{F}_{n,\lambda}^{-1}(p)d\psi(p),$$

*for any measure* $\psi$ *that may not have a density (e.g.,* $\psi$ *can be a point mass at* $p_0 \in (0,1)$*). Suppose*

1. $r_i \in [a, b]$ *for some* $-\infty < a < b < \infty$ *almost surely, with* $\inf_{r \in [a,b]} f_\lambda(r) > 0$, *and* $\sup_{r \in [a,b]} f'_\lambda(r) < \infty$.

2. $\int \frac{d\psi(t)}{f_\lambda(F_\lambda^{-1}(t))} < \infty$.

*then* $\sqrt{n}(\hat{R}_\psi(\lambda) - R_\psi(\lambda)) \xrightarrow{d} N(0, \sigma^2(\lambda))$ *where*

$$\sigma^2(\lambda) = \int_0^1 \int_0^1 \frac{p \wedge p' - p \cdot p'}{f_\lambda(F_\lambda^{-1}(p))f_\lambda(F_\lambda^{-1}(p'))}d\psi(p)d\psi(p') < \infty.$$

*Proof.* By the Bahadur-Kiefer representation (Kiefer, 1970) (see also Theorem E of (Csörgő & Révész, 1978)) of sample quantiles we can write

$$\Delta_n \triangleq \sup_{p \in [0,1]} |(F_{n,\lambda}^{-1}(p) - F_\lambda^{-1}(p))f_\lambda(F_\lambda^{-1}(p)) - (F_{n,\lambda}(F_\lambda^{-1}(p)) - p)|$$

$$= O_p\left(\frac{\log(n)^{1/2}(\log\log n)^{1/4}}{n^{3/4}}\right) = O_p\left(\frac{\log(n)}{n^{3/4}}\right),$$

equivalently,

$$\left|(\hat{F}_{n,\lambda}^{-1}(p) - F_\lambda^{-1}(p)) - \frac{F_{n,\lambda}(F_\lambda^{-1}(p)) - p}{f_\lambda(F_\lambda^{-1}(p))}\right| = \frac{\Delta_n}{f_\lambda(F_\lambda^{-1}(p))}.$$

Given this, we have that

$$\hat{R}_\psi(\lambda) - R_\psi(\lambda) = \int_0^1 (\hat{F}_{n,\lambda}^{-1}(p) - F_\lambda^{-1}(p))d\psi(p)$$

$$= \int_0^1 \frac{F_{n,\lambda}(F_\lambda^{-1}(p)) - p}{f_\lambda(F_\lambda^{-1}(p))}d\psi(p) + \delta_n$$

where

$$|\delta_n| \leq \Delta_n \cdot \int_0^1 \frac{1}{f_\lambda(F_\lambda^{-1}(p))}d\psi(p) = O_p\left(\frac{\log(n)}{n^{3/4}}\right)$$

holds by *assumption 2*.

Thus we have that

$$\sqrt{n}(\hat{R}_\psi(\lambda) - R_\psi(\lambda)) = \sqrt{n} \cdot \frac{1}{n}\sum_{i=1}^n \int_0^1 \frac{\mathbb{1}\{r_i \leq F_\lambda^{-1}(p)\} - p}{f_\lambda(F_\lambda^{-1}(p))}d\psi(p) + o_p(1)$$

$$\triangleq \frac{1}{\sqrt{n}}\sum_{i=1}^n S(r_i) + o_p(1)$$

By *assumption 2*,

$$S(r_i) = \int_0^1 \frac{\mathbb{1}\{r_i \leq F_\lambda^{-1}(p)\} - p}{f_\lambda(F_\lambda^{-1}(p))}d\psi(p) \leq \int_0^1 \frac{1}{f_\lambda(F_\lambda^{-1}(p))}d\psi(p) < \infty.$$

It remains to prove $\mathbb{E}[S(r_i)] = 0$ and $\text{Var}(S(r_i)) = \mathbb{E}[S^2(r_i)] < \infty$.

To study the mean, note that by *assumption 1*, $F_\lambda^{-1}(p)$ is uniquely defined and $\mathbb{P}(r_i \leq F_\lambda^{-1}(p)) = p$. Thus,

$$\mathbb{E}[S(r_i)] = \int_0^1 \frac{\mathbb{E}[\mathbb{1}\{r_i \leq F_\lambda^{-1}(p)\} - p]}{f_\lambda(F_\lambda^{-1}(p))}d\psi(p) = 0$$

To study the variance, observe that the second moment can be expressed as follows

$$\mathbb{E}[S^2(r_i)] = \mathbb{E}\left[\left(\int_0^1 \frac{\mathbb{1}\{r_i \leq F_\lambda^{-1}(p)\} - p}{f_\lambda(F_\lambda^{-1}(p))} d\psi(p)\right)^2\right]$$

$$= \mathbb{E}\left[\int_0^1 \frac{\mathbb{1}\{r_i \leq F_\lambda^{-1}(p)\} - p}{f_\lambda(F_\lambda^{-1}(p))} d\psi(p) \cdot \int_0^1 \frac{\mathbb{1}\{r_i \leq F_\lambda^{-1}(p')\} - p'}{f_\lambda(F_\lambda^{-1}(p'))} d\psi(p')\right]$$

$$= \int_0^1 \int_0^1 \frac{\mathbb{E}\left[(\mathbb{1}\{r_i \leq F_\lambda^{-1}(p)\} - p) \cdot (\mathbb{1}\{r_i \leq F_\lambda^{-1}(p')\} - p')\right]}{f_\lambda(F_\lambda^{-1}(p))f_\lambda(F_\lambda^{-1}(p'))} d\psi(p)d\psi(p')$$

Let $B(p) = \mathbb{1}\{r_i \leq F_\lambda^{-1}(p)\}$. Notice that $\mathbb{E}[B(p)] = p$, thus

$$\mathbb{E}\left[(B(p) - p) \cdot (B(p') - p')\right] = \text{Cov}(B(p), B(p'))$$

$$= \mathbb{E}\left[B(p) \cdot B(p')\right] - p \cdot p'$$

$$= \begin{cases} p - p \cdot p' & , \quad \text{if } p \leq p' \\ p' - p \cdot p' & , \quad \text{if } p > p' \end{cases}$$

$$= p \wedge p' - p \cdot p'$$

Therefore,

$$\text{Var}(S(r_i)) = \int_0^1 \int_0^1 \frac{p \wedge p' - p \cdot p'}{f_\lambda(F_\lambda^{-1}(p))f_\lambda(F_\lambda^{-1}(p'))} d\psi(p)d\psi(p')$$

By *assumption 2*,

$$\text{Var}(S(r_i)) \leq \int_0^1 \int_0^1 \frac{1}{f_\lambda(F_\lambda^{-1}(p))f_\lambda(F_\lambda^{-1}(p'))} d\psi(p)d\psi(p') = \left|\int_0^1 \frac{1}{f_\lambda(F_\lambda^{-1}(p))} d\psi(p)\right|^2 < \infty.$$

$\square$

**Theorem C.2** (Consistent variance estimate for L-Statistics). *In the same setting as Theorem C.1, assume the assumption 1 holds and that $\psi(y) = \int_0^y \psi'(z)dz$ for some $\psi'$ that is bounded and continuous at $F_\lambda(r)$ for Lebesgue almost-every $r$.*

*Let*

$$\hat{\sigma}_n^2(\lambda) = \int_a^b \int_a^b \psi'(\hat{F}_{n,\lambda}(r))\psi'(\hat{F}_{n,\lambda}(\tilde{r}))(\hat{F}_{n,\lambda}(r \wedge \tilde{r}) - \hat{F}_{n,\lambda}(r)\hat{F}_{n,\lambda}(\tilde{r})) \, dr \, d\tilde{r}.$$

*If $\psi'$ is bounded, then $\hat{\sigma}_n^2(\lambda) \xrightarrow{a.s.} \sigma^2(\lambda)$ as $n \to \infty$.*

*Proof.* We first verify *assumption 2*. Take $p = F_\lambda(x)$,

$$\int_0^1 \frac{1}{f_\lambda(F_\lambda^{-1}(p))} d\psi(p) = \int \frac{\psi'(p)}{f_\lambda(F_\lambda^{-1}(p))} dp$$

$$= \int_a^b \frac{\psi'(F_\lambda(x))}{f_\lambda(x)} dF_\lambda(x)$$

$$= \int_a^b \psi'(F_\lambda(x))dx.$$

The integral is bounded since $\psi'$ is bounded and $a, b$ are both finite. Let $p = F_\lambda(r)$, and $p' = F_\lambda(\tilde{r})$ in the double integral of $\sigma^2(\lambda)$, then

$$\sigma^2(\lambda) = \int_0^1 \int_0^1 \frac{p \wedge p' - p \cdot p'}{f_\lambda(F_\lambda^{-1}(p))f_\lambda(F_\lambda^{-1}(p'))} d\psi(p)d\psi(p')$$

$$= \int_a^b \int_a^b \frac{F_\lambda(r \wedge \tilde{r}) - F_\lambda(r)F_\lambda(\tilde{r})}{\cancel{f_\lambda(r)f_\lambda(\tilde{r})}} \psi'(F_\lambda(r))\psi'(F_\lambda(\tilde{r})) \cdot \cancel{f_\lambda(r)f_\lambda(\tilde{r})} \, dr \, d\tilde{r}.$$

By the Glivenko-Centelli Theorem, $\hat{F}_{n,\lambda}(r) \xrightarrow{a.s.} F_\lambda(r)$ for every $r \in [a, b]$, thus, for any $r, r' \in [a, b]$,

$$\hat{F}_{n,\lambda}(r \wedge \tilde{r}) - \hat{F}_{n,\lambda}(r)\hat{F}_{n,\lambda}(\tilde{r}) \xrightarrow{a.s.} F_\lambda(r \wedge \tilde{r}) - F_\lambda(r)F_\lambda(\tilde{r}).$$

Furthermore, since $\psi'$ is continuous at $F(r)$ for almost every $r$ under the Lebesgue measure, by the continuous mapping theorem,

$$\psi'(\hat{F}_{n,\lambda}(r)) \xrightarrow{a.s.} \psi'(F_\lambda(r))$$

for almost every $r$ under Lebesque measure. Suppose that $\psi'$ is bounded above by $B$, then

$$\psi'(\hat{F}_{n,\lambda}(r))\psi'(\hat{F}_{n,\lambda}(\tilde{r}))(\hat{F}_{n,\lambda}(r \wedge \tilde{r}) - \hat{F}_{n,\lambda}(r)\hat{F}_{n,\lambda}(\tilde{r})) \leq B^2,$$

thus

$$\hat{\sigma}_n^2(\lambda) \xrightarrow{a.s.} \sigma^2(\lambda)$$

by the Dominated Convergence Theorem.

$\square$

## C.2. Proof of Corollary 3.2

For CVaR$_\beta$, $\psi(p) = \max\{p - \beta, 0\}/(1 - \beta)$ and hence $\psi' = I(p \geq \beta)/(1 - \beta)$. By (C.2),

$$\sigma^2(\lambda) = \frac{1}{(1 - \beta)^2} \int_a^b \int_a^b (F_\lambda(r \wedge \tilde{r}) - F_\lambda(r)F_\lambda(\tilde{r}))I(F_\lambda(r) \geq \beta)I(F_\lambda(\tilde{r}) \geq \beta)drd\tilde{r}$$

$$= \frac{1}{(1 - \beta)^2} \int_{F_\lambda^{-1}(\beta)}^b \int_{F_\lambda^{-1}(\beta)}^b (F_\lambda(r \wedge \tilde{r}) - F_\lambda(r)F_\lambda(\tilde{r}))drd\tilde{r}.$$

Let $G_\lambda$ be the CDF of $r_i' \triangleq \max\{r_i, \beta\}$. Then

$$G_\lambda(r) = F_\lambda(r)I(r \geq F_\lambda^{-1}(\beta)).$$

By Hoeffding's covariance identity (Hoeffding, 1940),

$$\text{Var}(r_i') = \int_a^b \int_a^b (G_\lambda(r \wedge \tilde{r}) - G_\lambda(r)G_\lambda(\tilde{r}))drd\tilde{r} = \int_{F_\lambda^{-1}(\beta)}^b \int_{F_\lambda^{-1}(\beta)}^b (F_\lambda(r \wedge \tilde{r}) - F_\lambda(r)F_\lambda(\tilde{r}))drd\tilde{r}.$$

Thus,

$$\sigma^2(\lambda) = \frac{1}{(1 - \beta)^2}\text{Var}(r_i').$$

Since $r_i'$ is bounded, the Law of Large number implies that $\text{Var}(r_i')$ can be estimated consistently by the empirical variance of $(r_1', \ldots, r_n')$.

## C.3. Proof of Theorem 3.3

The proof is very similar to Bates et al. (2021), except that they only consider expected risk measures. If $R_\psi(\lambda) \leq \alpha$ for all $\lambda \in \Lambda$, then the result obviously holds. Assume $\sup_{\lambda \in \Lambda} R_\psi(\lambda) > \alpha$. Since $R_\psi(\lambda)$ is continuous and strictly increasing, it crosses $\alpha$ exactly once. Let $\lambda^*$ denote the crossing point, i.e., $R_\psi(\lambda^*) = \alpha$. Then

$$R_\psi(\hat{\lambda}) > \alpha \iff \hat{\lambda} > \lambda^* \implies \hat{R}_\psi^+(\lambda^*) \leq \alpha,$$

where the last line is due to the definition of $\hat{\lambda}$. By Theorem 3.1,

$$\lim_{n \to \infty} \mathbb{P}(\hat{R}_\psi^+(\lambda^*) < R_\psi(\lambda^*)) = \delta.$$

Since $R_\psi(\lambda^*) = \alpha$, we have

$$\limsup_{n \to \infty} \mathbb{P}(R_\psi(\hat{\lambda}) > \alpha) \geq \limsup_{n \to \infty} \mathbb{P}(\hat{R}_\psi^+(\lambda^*) < R_\psi(\lambda^*)) = \delta.$$

The proof is then completed.

# D. Distortion Risk Control for $\text{VaR}_\beta$

## D.1. Asymptotics of empirical quantiles

Take $\psi(q) = \delta_\beta(q)$, we obtain $\text{VaR}_\beta(\lambda) = F_\lambda^{-1}(\beta)$. Theorem 3.1 implies that the empirical $\beta$-th quantile is asymptotically normal with variance

$$\sigma^2_{\text{VaR}_\beta}(\lambda) = \frac{\beta(1-\beta)}{f_\lambda^2(F_\lambda^{-1}(\beta))}.$$

Empirically, we use the standard Bootstrap procedure to estimate $\sigma^2_{\text{VaR}_\beta}(\lambda)$ with 1000 samples.

## D.2. Additional experimental results for $\text{VaR}_\beta$

We evaluate the performance, i.e., average sampling cost, and realized $\text{VaR}_\beta$ on a held-out dataset with $|\mathcal{D}| = 6000$, of our framework CDRC-L, against baseline methods CDRC-DKW and CDRC-BJ. See Fig. 13 for $\rho = 0.68$, and Fig. 14 for $\rho = 0.78$. Similar to what we observe with $\text{CVaR}_\beta$, it is evident from the plots that all methods control the risk at the specified level $\alpha$ since they fall below the black dotted line (where $\alpha = \text{VaR}_\beta$) within the margin of error. Again, our method, CDRC-L, is consistently the least conservative of all methods across different values of $\beta$, and $\rho$. As a result, we see that our method is consistently the least costly across all settings of $\beta$, and $\rho$. Fig. 15 illustrates that for all settings of $\beta$, as the Spearman correlation coefficient increases, the cost of generating a $\text{VaR}_\beta$-controlled LLM response decreases.

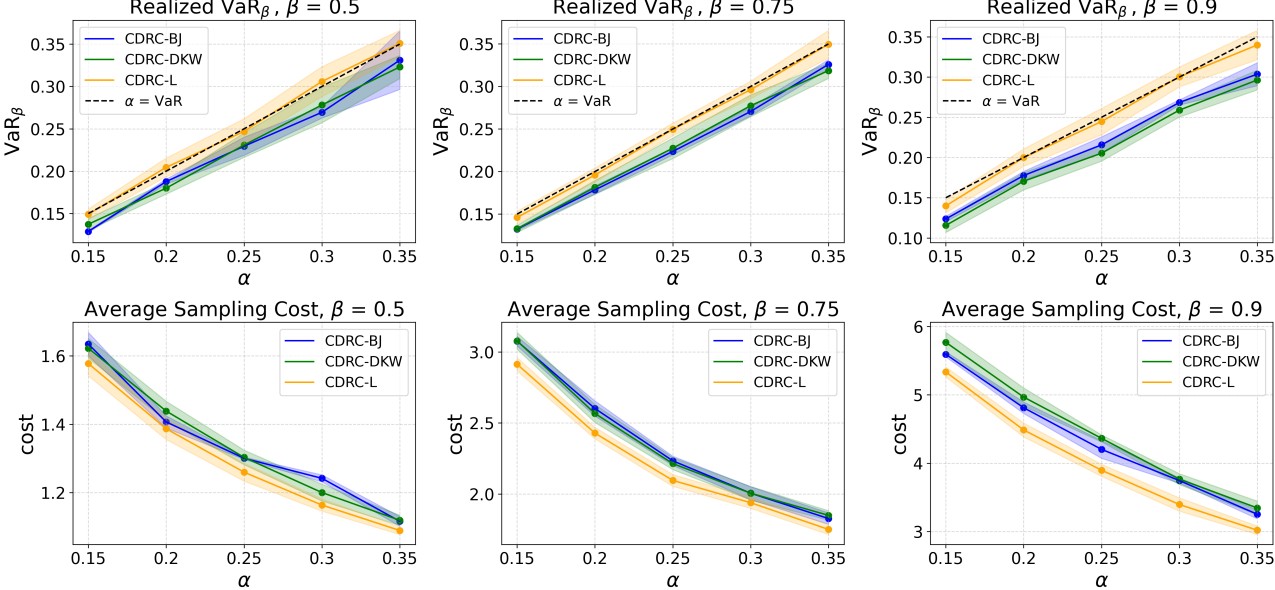

*Figure 12.* **Realized $\text{VaR}_\beta$ vs. $\alpha$ (row 1) and average sampling cost vs. $\alpha$ (row 2), and for Spearman correlation between human and machine toxicity scores at $\rho = 0.57$ evaluated on held-out dataset with $|\mathcal{D}| = 6000$.** The confidence band is computed by taking the mean estimate plus/minus one standard error estimated from the results across the independent experiments. Each panel in the respective rows illustrates a different setting of $\beta \in \{0.5, 0.75, 0.9\}$. CDRC-L (orange), is an improvement to CDRC-DKW (green) and CDRC-BJ (blue), as it is able to achieve risk control within the margin of error (shown in the black dotted line) while being more cost efficient and less conservative than both baseline methods.

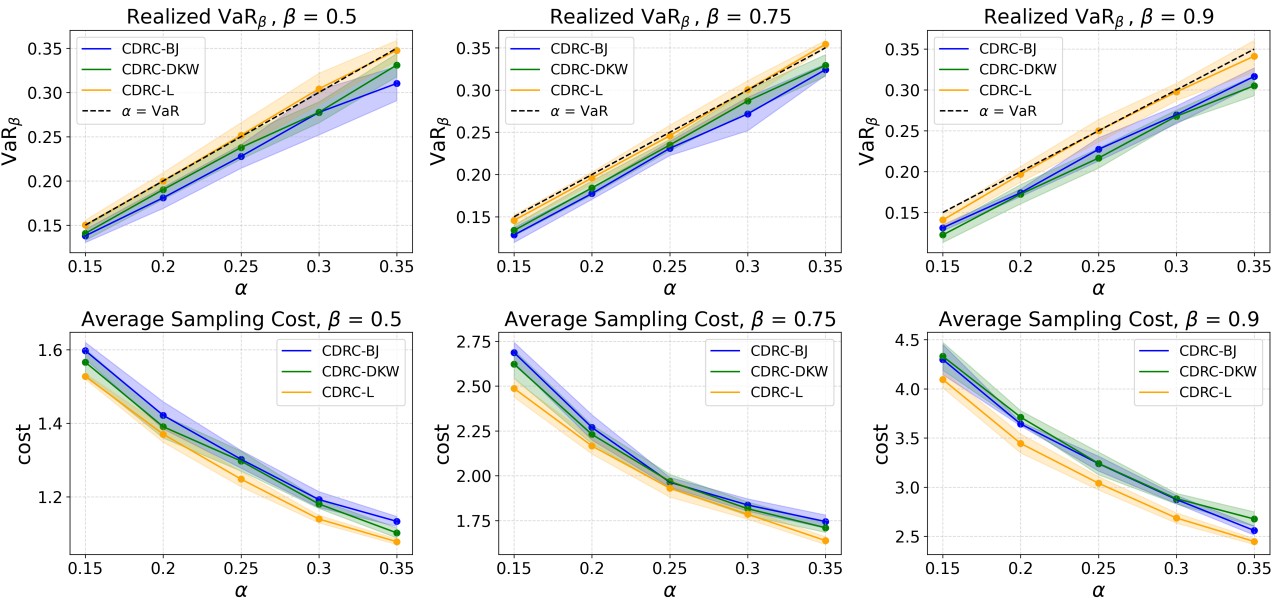

*Figure 13.* **Average cost and realized** $\mathrm{VaR}_\beta$ **on held-out dataset for** $\rho = 0.68$ **with** $|\mathcal{D}| = 6000$.

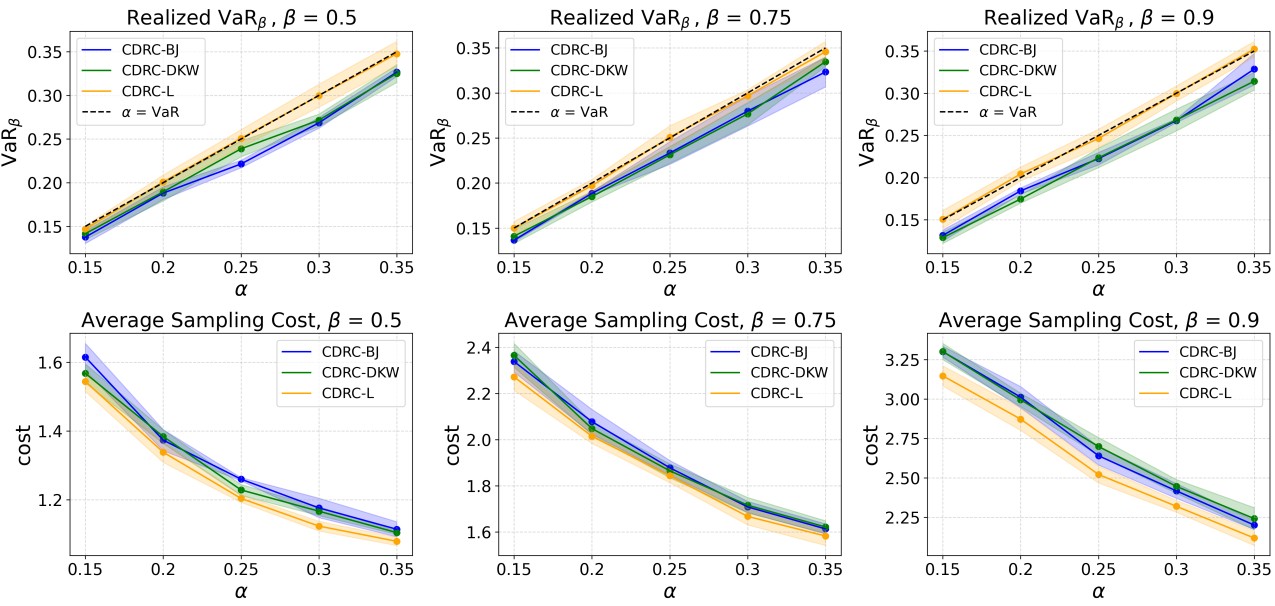

*Figure 14.* **Average cost and realized** $\mathrm{VaR}_\beta$ **on held-out dataset for** $\rho = 0.78$ **with** $|\mathcal{D}| = 6000$.

Similar to the results for $\mathrm{CVaR}_\beta$, from Fig. 16, it is evident that CDRC-L is consistently the least conservative among all methods, while still maintaining the guaranteed $\mathrm{VaR}_\beta$ control within the margin of error. Furthermore, from Fig. 17, we observe that CDRC-L is consistently the most cost effective among all methods. In particular, it is significantly less expensive in comparison to other methods in settings with small calibration set sizes.

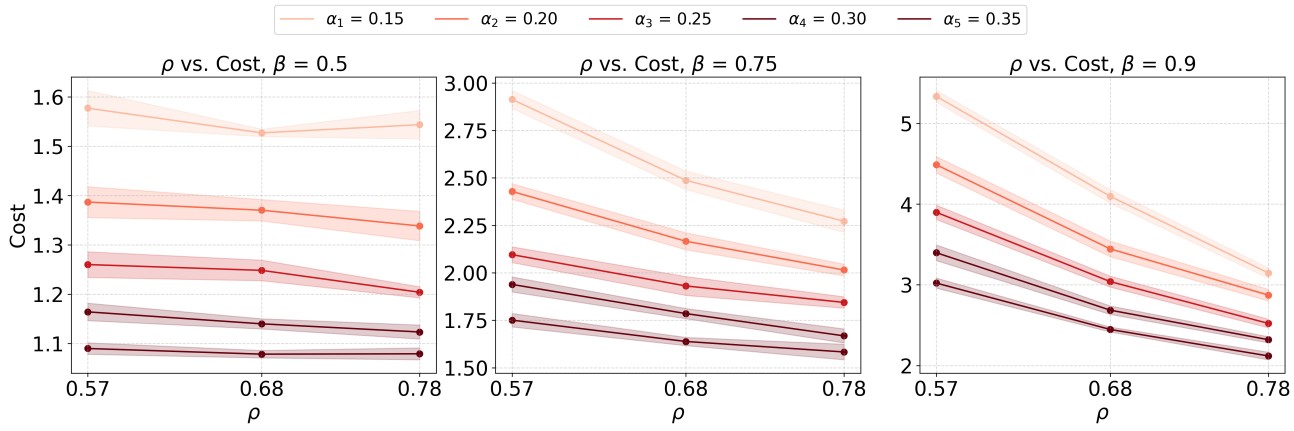

*Figure 15.* **Spearman correlation vs. average sampling cost for** $\mathrm{VaR}_\beta$ **with** $|\mathcal{D}| = 6000$.

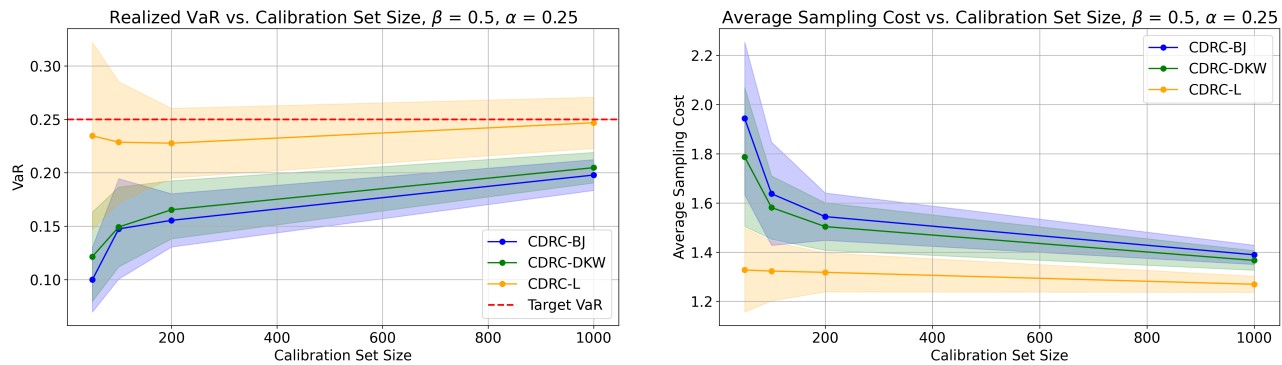

*Figure 16.* **Realized** $\mathrm{VaR}_\beta$ **vs. calibration set size** for $\beta = 0.5$, $\alpha = 0.25$, $\rho = 0.57$.

*Figure 17.* **Average sampling cost for** $\mathrm{VaR}_\beta$ **control vs. calibration set size** for $\beta = 0.5$, $\alpha = 0.25$, $\rho = 0.57$.

# E. Additional model results

## E.1. LLaMA3.1-8B

To evaluate our tail risk control framework using the LlaMA3.1-8B model, we choose $\alpha \in \{0.125, 0.15, 0.2, 0.25, 0.3\}$ for $\beta \in \{0.5, 0.75\}$, and $\alpha \in \{0.2, 0.25, 0.3\}$ for $\beta = 0.9$ for testing $\mathrm{CVaR}_\beta$ control. In addition, we choose $\alpha \in \{0.05, 0.075, 0.1\}$ for $\beta = 0.5$, and $\alpha \in \{0.1, 0.15, 0.2, 0.25, 0.3\}$ for $\beta \in \{0.75, 0.9\}$ for testing $\mathrm{VaR}_\beta$ control. Using $\rho = 0.57$, we illustrate the results for $\mathrm{CVaR}_\beta$ and $\mathrm{VaR}_\beta$ control in 18 and 19, respectively for $|\mathcal{D}| = 6000$.

We examine the performance, i.e. the realized $\mathrm{CVaR}_\beta$ and $\mathrm{VaR}_\beta$, and average sampling cost of our framework, CDRC-L, on a held-out dataset against baseline methods CDRC-DKW and CDRC-BJ. Similar to what we observe with the LlaMA2-7B model, it is evident from the plots that all methods control the risk at the specified level $\alpha$ since they fall below the black dotted line (where $\alpha = \mathrm{CVaR}_\beta$ or $\mathrm{VaR}_\beta$) within the margin of error. Meanwhile, our method, CDRC-L, is consistently the least conservative and hence most cost efficient of all methods across different values of $\beta$.

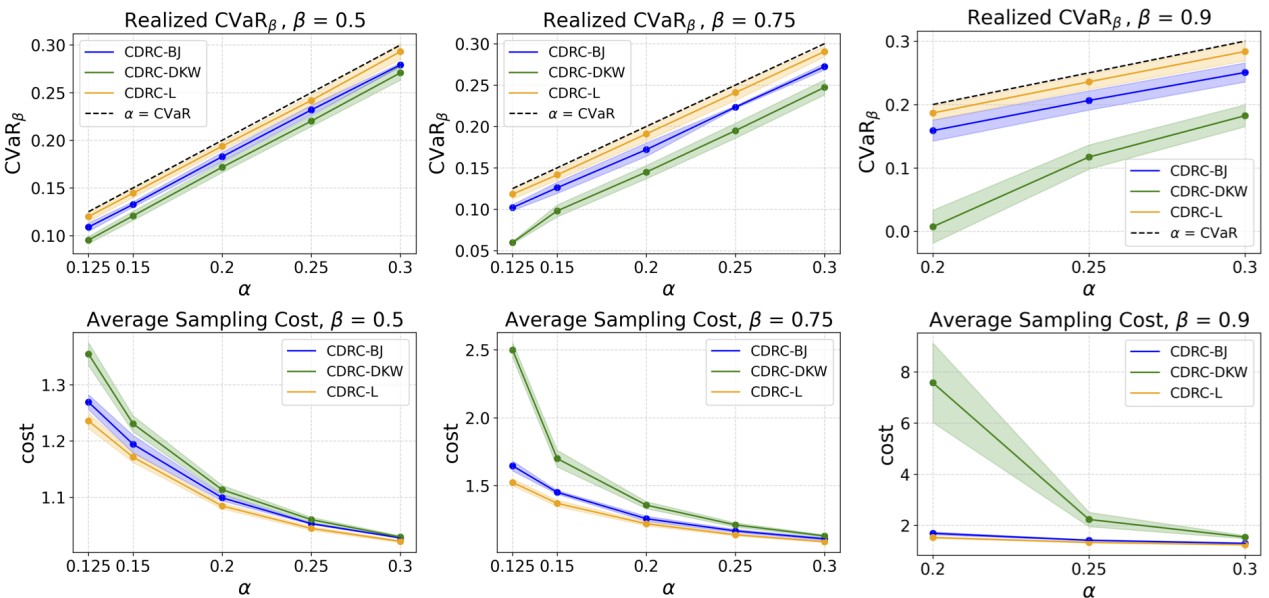

*Figure 18.* **LLaMA3.1-8B:Realized** $\text{CVaR}_\beta$ **vs.** $\alpha$ **(row 1), and average sampling cost vs.** $\alpha$ **(row 2) for Spearman correlation between human and machine toxicity scores at** $\rho = 0.57$ **evaluated on held-out dataset with** $|\mathcal{D}| = 6000$. Each panel in the respective rows illustrates a different setting of $\beta \in \{0.5, 0.75, 0.9\}$. CDRC-L (orange), is an improvement to CDRC-DKW (green) and CDRC-BJ (blue), as it is able to achieve risk control within the margin of error (shown in the black dotted line) while being more cost efficient and less conservative than both baseline methods.

## E.2. Llama3.2-3B

To evaluate our tail risk control framework using the LlaMA3.2-3B model, we choose $\alpha \in \{0.15, 0.2, 0.25, 0.3, 0.35\}$ for $\beta \in \{0.5, 0.75\}$, and $\alpha \in \{0.2, 0.25, 0.3\}$ for $\beta = 0.9$ for testing $\text{CVaR}_\beta$ control. In addition, we choose $\alpha \in \{0.15, 0.2, 0.25, 0.3, 0.35\}$ for $\beta \in \{0.5, 0.75, 0.9\}$ for testing $\text{VaR}_\beta$ control. Using $\rho = 0.57$, we illustrate the results for $\text{CVaR}_\beta$ and $\text{VaR}_\beta$ control in 20 and 21, respectively for $|\mathcal{D}| = 6000$.

Again, we examine the performance, i.e. the realized $\text{CVaR}_\beta$ and $\text{VaR}_\beta$, and average sampling cost of our framework, CDRC-L, on a held-out dataset against baseline methods CDRC-DKW and CDRC-BJ. Similar to what we observe with the LlaMA2-7B and LlaMA3.1-8B model, it is evident from the plots that all methods control the risk at the specified level $\alpha$ since they fall below the black dotted line (where $\alpha = \text{CVaR}_\beta$ or $\text{VaR}_\beta$) within the margin of error. Meanwhile, our method, CDRC-L, is consistently the least conservative and therefore, the most least costly of all methods across different values of $\beta$.

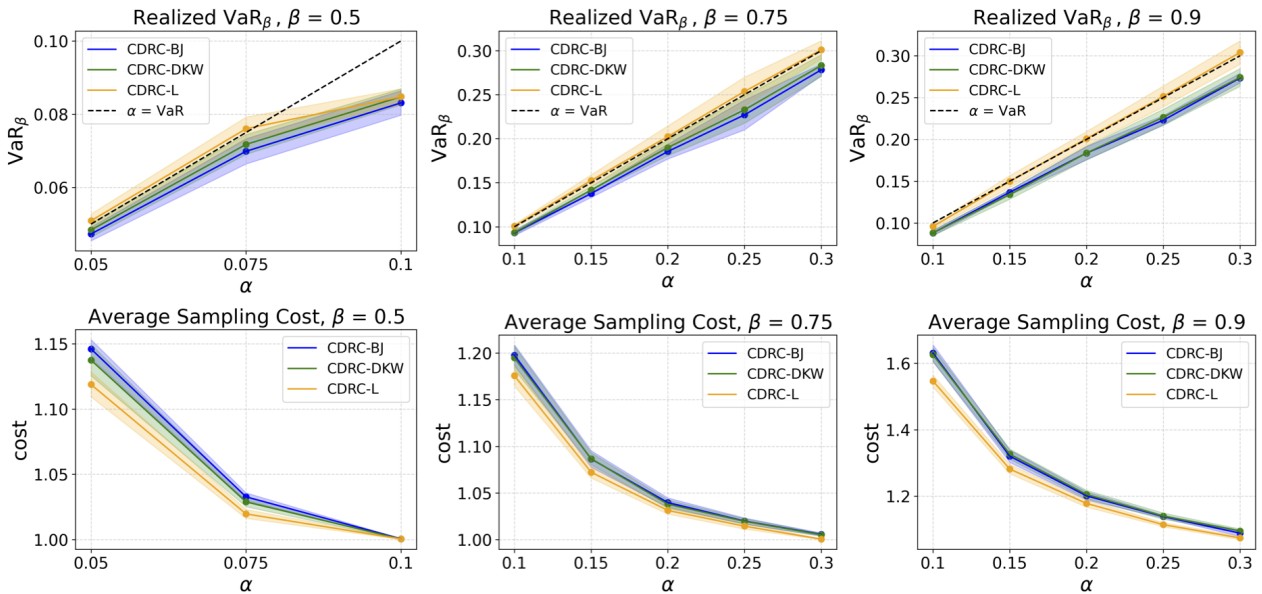

*Figure 19.* **LLaMA3.1-8B: Realized** $\mathrm{VaR}_\beta$ **vs.** $\alpha$ **(row 1), and average sampling cost vs.** $\alpha$ **(row 2), and for Spearman correlation between human and machine toxicity scores at** $\rho = 0.57$ **evaluated on held-out dataset with** $|\mathcal{D}| = 6000$. Each panel in the respective rows illustrates a different setting of $\beta \in \{0.5, 0.75, 0.9\}$. CDRC-L (orange), is an improvement to CDRC-DKW (green) and CDRC-BJ (blue), as it is able to achieve risk control within the margin of error (shown in the black dotted line) while being more cost efficient and less conservative than both baseline methods.

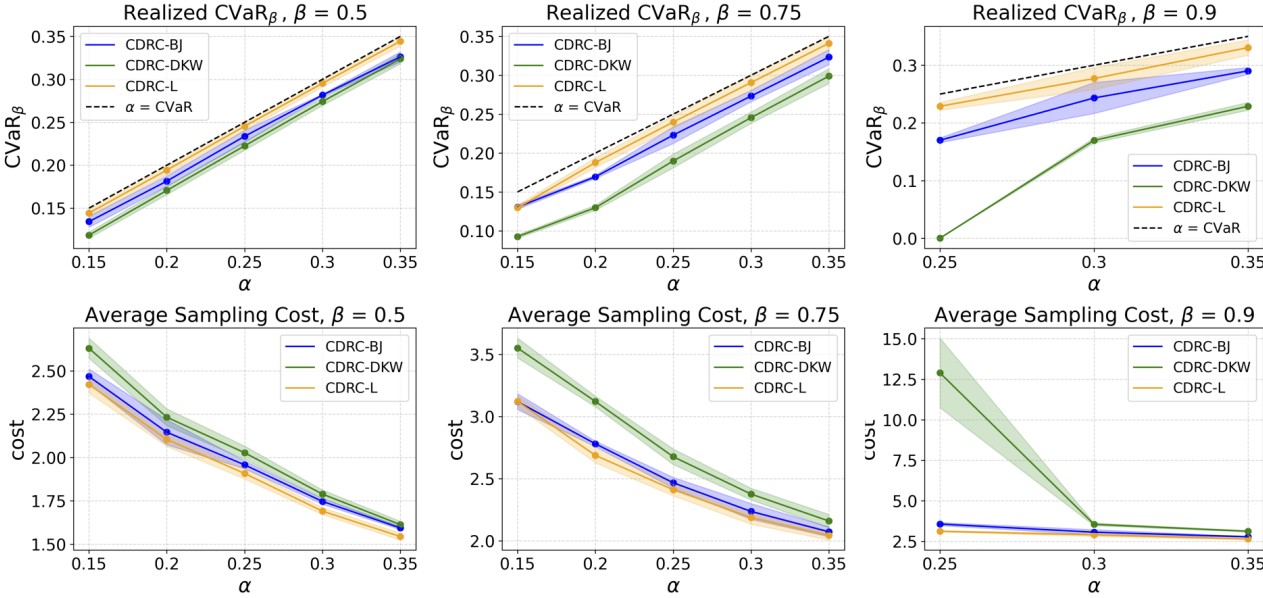

*Figure 20.* **Llama3.2-3B: Realized** $\mathrm{CVaR}_\beta$ **vs.** $\alpha$ **(row 1), and average sampling cost vs.** $\alpha$ **(row 2) for Spearman correlation between human and machine toxicity scores at** $\rho = 0.57$ **evaluated on held-out dataset with** $|\mathcal{D}| = 6000$. Each panel in the respective rows illustrates a different setting of $\beta \in \{0.5, 0.75, 0.9\}$. CDRC-L (orange), is an improvement to CDRC-DKW (green) and CDRC-BJ (blue), as it is able to achieve risk control within the margin of error (shown in the black dotted line) while being more cost efficient and less conservative than both baseline methods.

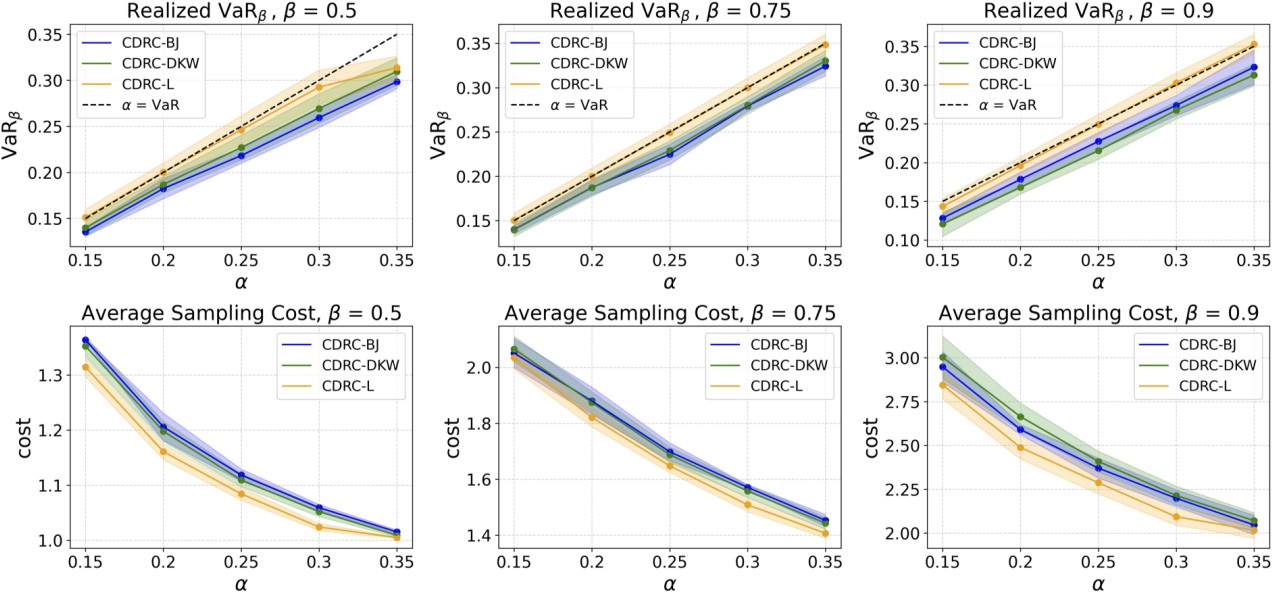

*Figure 21.* **Llama3.2-3B: Realized** $\mathrm{VaR}_\beta$ **vs.** $\alpha$ **(row 1), and average sampling cost vs.** $\alpha$ **(row 2) for Spearman correlation between human and machine toxicity scores at** $\rho = 0.57$ **evaluated on held-out dataset with** $|\mathcal{D}| = 6000$. Each panel in the respective rows illustrates a different setting of $\beta \in \{0.5, 0.75, 0.9\}$. CDRC-L (orange), is an improvement to CDRC-DKW (green) and CDRC-BJ (blue), as it is able to achieve risk control within the margin of error (shown in the black dotted line) while being more cost efficient and less conservative than both baseline methods.

